# Spatiotemporal distribution analysis of syphilis in Brazil: Cases of congenital and syphilis in pregnant women from 2001–2017

**Ângelo Antônio Oliveira Silva**[1,2], **Leonardo Maia Leony**[1], **Wayner Vieira de Souza**[3], **Natália Erdens Maron Freitas**[1], **Ramona Tavares Daltro**[1], **Emily Ferreira Santos**[1], **Larissa de Carvalho Medrado Vasconcelos**[1], **Maria Fernanda Rios Grassi**[1,4], **Carlos Gustavo Regis-Silva**[1], **Fred Luciano Neves Santos**[1,5]*

1 Advanced Public Health laboratory, Gonçalo Moniz Institute, Oswaldo Cruz Foundation (Fiocruz-BA), Salvador, Bahia, Brazil, 2 Salvador University (UNIFACS), Salvador, Bahia, Brazil, 3 Department of Public Health, Aggeu Magalhães Institute, Oswaldo Cruz Foundation (Fiocruz-PE), Recife, Pernambuco, Brazil, 4 Bahiana School of Medicine and Public Health (EBMSP), Salvador, Bahia, Brazil, 5 Integrated Translational Program in Chagas Disease from Fiocruz (Fio-Chagas), Oswaldo Cruz Foundation (Fiocruz-RJ), Rio de Janeiro, Rio de Janeiro, Brazil

* fred.santos@fiocruz.br

**Data Availability Statement:** All relevant data are within the manuscript and its Supporting information files.

## Abstract

In Brazil, the notification of congenital (CS) and syphilis in pregnant women (SiP) is compulsory. Notification data provided by the Ministry of Health in combination with the mapping of vulnerable geographic areas is essential to forecasting possible outbreaks and more effectively combating infection through monitoring. We aim to evaluate the spatiotemporal distribution and epidemiological aspects of reported cases of CS and SiP in Brazil. A retrospective ecological study was carried out using secondary surveillance data obtained from the Brazilian National Notifiable Diseases Information System (SINAN) database, considering all reported cases of CS and SiP between 2001 to 2017. Epidemiological characteristics and time trends were analyzed using joinpoint regression models and spatial distribution, considering microregions or states/macroregions as units of analysis. A total of 188,630 (359/100,000 birth lives) CS and 235,895 of SiP (6.3/100,000 inhabitants) were reported during the period studied. In general, the epidemiologic profile of Brazil indicates most reported CS cases occurred in "mixed-race" newborns who were diagnosed within seven days of birth and whose mothers had received prenatal care, but the epidemiologic profile varies by Brazilian macroregion. Regarding SiP, most cases were among women who self-reported 'mixed-race', were aged 20–39 years, had up to eight years of formal education and were diagnosed with primary or latent syphilis. Approximately 549 (98.4%) and 558 (100%) microregions reported at least one case of CS and SiP, respectively. From 2012 to 2016, CS cases increased significantly in almost all Brazilian states, most notably in the South, Southeast, and Central-West macroregions, from 2001–2017 and the relative risk (RR) of SiP increased around 400% (RR: 1,00 to 445,50). Considering the epidemiological scenario of the infection in Brazil, it is necessary to enhance preventive, control and eradication measures.

**Funding:** This work was supported by the Coordination of Superior Level Staff Improvement-Brazil (CAPES; Finance Code 001) and the Research Support Foundation of the State of Bahia (Fundação de Amparo à Pesquisa do Estado da Bahia - FAPESB; BOL0261/2018 and BOL0932/20). Wayner Vieira de Souza, Maria Fernanda R. Grassi, and Fred Luciano N. Santos are research fellows of the Brazilian National Council for Scientific and Technological Development (Conselho Nacional de Desenvolvimento Científico e Tecnológico - CNPq; process no. 306222/2013-2, 304811/2017-3, and 309263/2020-4, respectively). The funders had no role in study design, data collection and analysis, decision to publish, or preparation of the manuscript.

**Competing interests:** The authors declare that they have no competing interests.

## Introduction

Syphilis, a sexually transmitted infection (STI), is a persistent public health issue [1]. According to the number of reported cases of curable STIs in 2016, only 6.3 million (1.67%) were syphilis in women and men aged 15–49 years. Looking at the period from 2012 to 2016, the estimated global prevalence was 0.5% and the incidence was estimated at 1.7 cases per 1,000 women and 1.6 cases per 1,000 men [2]. In 2016, the estimated global prevalence of syphilis in pregnant women was 0.69%, resulting in a global congenital syphilis rate of 473 cases per 100,000 live births (~661,000 total cases). The data showed that maternal syphilis caused 143,000 early fetal deaths and stillbirths, 61,000 neonatal deaths, 41,000 preterm or low birth weight births, 109,000 infants with clinical congenital syphilis, and 306,000 cases of infants without clinical signs in mothers with untreated syphilis [3]. In Brazil, 49,013 cases of syphilis in pregnant women (SiP) and 24,666 cases of congenital syphilis (CS) were notified in 2017 [4].

Syphilis, a systemic infection exclusive to humans caused by the bacterium *Treponema pallidum* subsp. *pallidum*, is mainly transmitted through unprotected intercourse (acquired syphilis) [1, 5]. Less commonly, non-sexual transmission can occur through treponemal lesions or by exposure to blood or contaminated body fluids [6–9]. In vertical transmission from mother-to-child (CS), the infection spreads to the fetus hematologically, predominantly via the transplacental route [10, 11]. Most of the signs and symptoms of the disease, e.g. tissue damage, arise from the inflammatory reaction to infection [12].

Despite the existence of diagnostic tests and effective antibiotic treatment, increasing numbers of syphilis cases registered in Brazil reflect the fragility of the public health system. In accordance with Brazilian Health Regulations (Ordinances 542 for CS– 1986; and 33 for SiP– 2005), syphilis infection requires compulsory notification to contribute to incidence/epidemiological investigations. The notifications sent to public health authorities provide data to the Ministry of Health (MoH) for monitoring and to predict potential outbreaks [13]. While the relationship between infections in specific groups has already been well established, the mapping of vulnerable geographic areas is essential to more effectively combat infection [4].

According to the Brazil MoH, women diagnosed with syphilis during pregnancy, at delivery and / or puerperium should be reported as SiP. This case definition includes symptomatic or asymptomatic pregnant women with at least one reactive syphilis test, either treponemal or nontreponemal (of any titer), and without previous recorded syphilis treatment. The case definition of CS includes: all newborns, stillbirths or abortions from women diagnosed with syphilis that have not been treated or who received inadequate treatment, children under 13 years of age with clinical manifestations, radiographic or radiological alterations and reactive nontreponemal or treponemal syphilis tests, and children, products of abortion or stillbirth with biopsy or necropsy microbiological evidence of *T. pallidum* infection in a sample of nasal discharge or skin lesion, detection of *T. pallidum* by means of direct exams by microscopy (dark field or with colored material) [14].

Globally, an increasing number of syphilis cases have also been reported in Africa, the Americas, the Eastern Mediterranean, Europe, Southeast Asia, and the Western Pacific [2, 3], which has drawn attention to a public health crisis related to the synergy between the epidemiology and biology of syphilis and the human immunodeficiency virus [15]. In Latin America, Asia and Africa, recent increases in case frequency and the size of affected geographic areas have marked a new era of infection transmission, seriously burdening limited-resource public health systems [16]. Studies on the incidence of infectious disease in terms of spatial distribution have proven important for public health analysis, especially with respect to the planning and execution of measures aimed at controlling disease. The present study evaluated the

spatiotemporal distribution and epidemiological aspects of reported cases of congenital and syphilis infection in pregnancy in Brazil between the years of 2001 and 2017.

## Materials and methods

### Ethics

As this study was based on secondary data, and all presented information is in the public domain, none of the described variables allowed for individual identification. In 2016, a new resolution published by the Brazilian National Health Council abrogated the need to seek approval from any Institutional Review Board for studies using publicly available secondary data that does not provide individually identifiable information (http://conselho.saude.gov.br/resolucoes/2016/reso510.pdf).

### Study area

This study was performed in Brazil, the fifth largest country in the world, with a total area of 8,515,767 km$^2$. The Federative Republic of Brazil is comprised of 26 states and a Federal District, which were grouped into five macroregions (Central-West, North, Northeast, South, and Southeast–see Fig 1), 137 mesoregions and 558 microregions by the Brazilian Institute of Geography and Statistics (IBGE) according to economic and social similarities. According to the 2015 Brazilian national census, the country's total population size was 204,338,473 inhabitants, with an overall density of 24 inhabitants per km$^2$ (data publicly available at http://www.ibge.gov.br).

### Study design

A retrospective ecological study was carried out using data from the open-access Brazilian National Notifiable Diseases Information System (SINAN) database, considering all reported cases of CS and SiP between 2001 to 2017 (data available at http://datasus.saude.gov.br). All cases reported by the 5,567 Brazilian municipalities were included in this study, and then grouped in the 558 microregions established by IBGE. Other variables were also analyzed in this study according to the classification of syphilis notification. With respect to SiP, self-reported skin color, age group, clinical stages of syphilis and years spent in education were considered. For CS, the variables of sex, self-reported skin color, age group and performing or not prenatal care were analyzed. Self-reported skin color was classified as European ancestry, dark skin, East Asian ancestry, indigenous ancestry, and mixed-race (persons whose skin color is not classified as dark-skinned, European, indigenous, or East Asian).

### Statistical analysis

The distribution of cases of CS and SiP were evaluated using spatial analysis methods and geo-processing techniques. Data on CS and SiP were analyzed according to the microregion in which notification occurred to allow for greater precision in intra- and inter-regional differences, and also to aid in revealing specific areas in which to target potential intervention efforts. To minimize potential interference from random fluctuations that could occur in time-series studies, distribution maps were constructed using three-year moving averages from 2001 to 2017; annual incidence was calculated for 558 Brazilian microregions. The infection rates for CS and SiP were expressed as the number of infected individuals per 100,000 live births and 100,000 inhabitants, respectively, and used to construct thematic maps. Temporal changes in annual incidence rates were calculated using the joinpoint regression model and expressed as Annual Percentage Change (APC) with 5% significance ($p < 0.05$) using the NCI

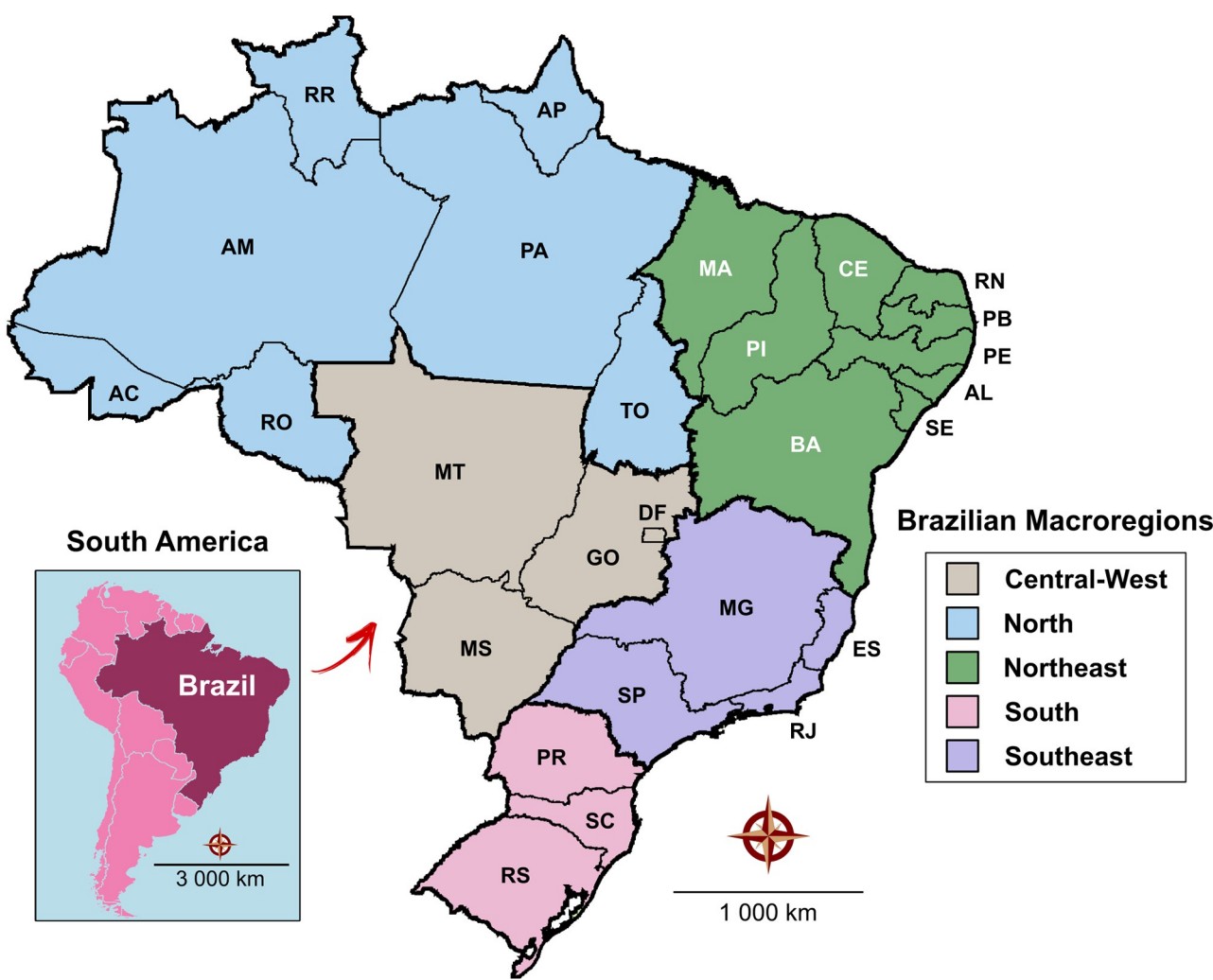

**Fig 1. Geographic division of Brazil into five macroregions, 26 states and a Federal District (DF).** Macroregions and corresponding state abbreviations: North (AC: Acre, AM: Amazonas, AP: Amapá, RO: Rondônia and RR: Roraima); Northeast (AL: Alagoas, BA: Bahia, CE: Ceará, MA: Maranhão, PB: Paraíba, PE: Pernambuco, PI: Piauí, RN: Rio Grande do Norte and SE: Sergipe); Central-West (DF: Distrito Federal, GO: Goiás, MS: Mato Grosso do Sul and MT: Mato Grosso); Southeast (ES: Espírito Santo, MG: Minas Gerais, RJ: Rio de Janeiro, and SP: São Paulo); South (PR: Paraná, RS: Rio Grande do Sul and SC: Santa Catarina). Digital maps in the public domain (publicly available) were obtained from IBGE cartographic database in shapefile format (.shp), which was subsequently reformatted and analyzed using QGIS version 3.10. Authors specify that this figure is licensed under CC BY 4.0.

Joinpoint regression program version 4.1.1 [17, 18]. To determine the optimal number of join-points, sequential permutation tests were performed during model selection. Each of the permutation tests performs a test of the null hypothesis H0: number of joinpoints = ka against the alternative Ha: number of joinpoints = kb with ka < kb. The procedure starts with ka = MIN or minimum number of joinpoints and kb = MAX or maximum number of joinpoints, in our case 0 and 5, respectively. Monte Carlo simulation, with the number of permutations fixed at 4,499, is used to calculate the permutation p-value for each hypothesis test. Based on the Joinpoint Regression Program recommendations for the number of time points of observations in our study, our analyzes allowed for a maximum of five joinpoints, meaning that between one and six trend segments could be included in the final model, depending on the number of joinpoints detected [19]. Publicly available digital maps were obtained from the IBGE cartographic

database in shapefile (.shp) format, then reformatted and analyzed using QGIS version 3.10 (Geographic 140 Information System, Open-Source Geospatial Foundation Project. http://qgis.osgeo.org). This software package was used for data processing, analysis, and the presentation of cartographic data. A checklist (S1 Checklist) is provided according to the Strengthening the Reporting of Observational studies in Epidemiology (STROBE) guidelines.

## Results

### Annual Percentage Change (APC) in syphilis cases

Between 2001 and 2017, 188,630 cases of CS and 235,895 cases of SiP were reported in Brazil. The mean overall rate of CS was 359 cases per 100,000 live births during the period studied, ranging from 143 to 891, while the mean rate per 100,000 inhabitants of SiP was 6.3, ranging from 0.02 to 19.1. From 2002 to 2016, joinpoint regression analysis revealed a generally increasing trend (Fig 2), highlighting two distinct and statistically significant epidemiological periods for CS and three for SiP. With respect to CS (Fig 2A), an average annual incidence rate of 184 cases per 100,000 live births was seen in the first period (2002 to 2009; APC = 5.1 [95% CI: 2.1; 8.2], Z = 3.8, p = 0.003), which increased to 514 in the second period (2009 to 2016; APC = 23.7 [95%CI: 20.1; 27.3], Z = 16.3, p < 0.001). Considering SiP (Fig 2B), an average annual incidence rate of 0.6 cases per 100,000 inhabitants was observed in the first period (2002 to 2006; APC = 203.26 [95%CI: 81.4; 406.9], Z = 5.1, p <0.01), which increased to 6.5 per 100,000 inhabitants (2006 to 2014; APC = 27.50 [95%CI: 23.8; 31.3], Z = 19.4, p < 0.01) in the second period. In the third period, the number of cases increased dramatically to 16.6 cases per 100,000 inhabitants (APC = 16.47 [95%CI: 3.6; 30.9], Z = 3.6, p < 0.01). Periods 1 and 2 of the epidemiological periods identified by joinpoint regression in both CS and SiP were not similar in terms of length or specific years.

### Sociodemographic profile of syphilis in Brazil

Fig 3 (S1 Table) illustrates changes in the profiles of the sociodemographic variables of CS cases during the two periods identified. Considering racial classification, the North, Northeast and Central-West macroregions reported more cases of CS among children who were identified as 'mixed-race' in the two periods: P1 (64.3%) and P2 (80.2%); P1 (51.2%) and P2 (68.5%), and P1 (29.6%) and P2 (48.1%), respectively. In the Southeast macroregion, more CS cases (25.9%) were identified among children identified as 'European ancestry' in P1, while in P2 (37.7%) more cases were identified as 'mixed-race'. Conversely, in the South macroregion, CS was predominant among children identified as 'European ancestry' in P1 (59.7%) and P2 (66.4%). With respect to age, more than 88% of the reported cases were diagnosed among children less than 7 days old in all periods and all macroregions. With respect to prenatal care among the CS cases, over 75% reported receiving prenatal care regardless of period and macroregion, suggesting a lack of prenatal screening. Interestingly, CS was generally reported in all macroregions at similar frequencies among males and females (about 50%).

Fig 4 (S2 Table) illustrates changes in the sociodemographic profile of SiP during the three distinct periods identified. With respect to self-reported skin color, the North (P1: 64.7%; P2: 63.7%; P3: 80.5%) and Northeast (P1: 64.5%; P2: 63.7%; P3: 66.6%) macroregions reported more cases of SiP in women who self-identified as 'mixed-race' in all periods, whereas the Central-West macroregion reported cases only in P2 (46.2%) and P3 (55.3%). Among women who self-identified as 'European ancestry', most cases were concentrated in the South (49.7%) and Southeast (43.7%) macroregions in P1, while 67.1% of cases were reported in the South macroregion in P3. With respect to age, more than 66% of cases were reported in women aged 20 to 39 years in all macroregions and all three periods. Regarding education level, most reported

## A - Congenital syphilis

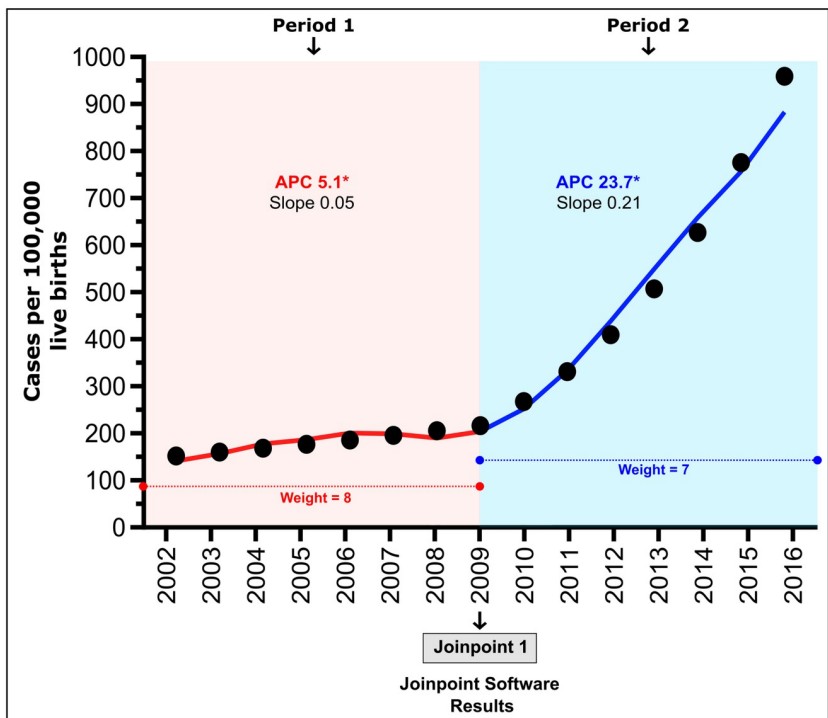

## B - Syphilis in pregnancy

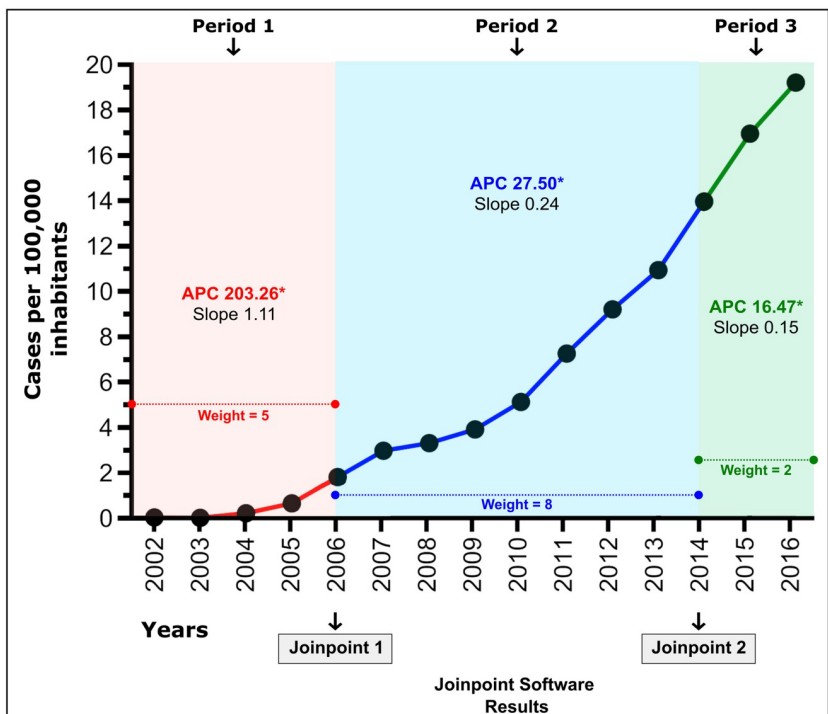

**Fig 2. Global incidence in Brazil per 100,000 inhabitants of CS (A) and SiP (B) according to SINAN.** Joinpoint regression analysis using APC (Annual Percentage Change) calculations identified three distinct periods for each form of syphilis.

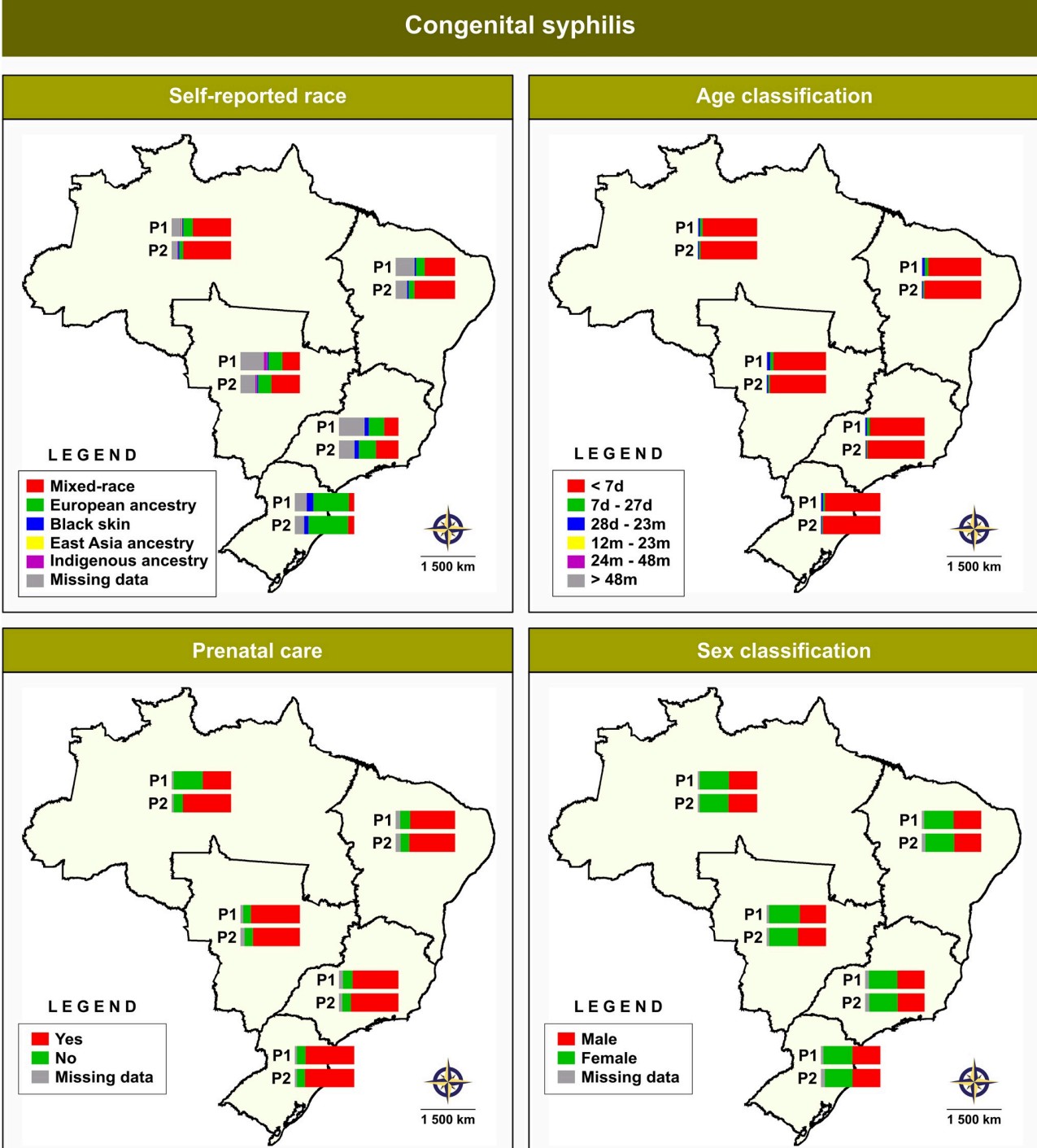

**Fig 3. Analysis of changes in sociodemographic variables by region, stratified according to two periods of CS notifications.** Digital maps in the public domain (publicly available) were obtained from IBGE cartographic database in shapefile format (.shp), which was subsequently reformatted and analyzed using QGIS version 3.10. Authors specify that this figure is licensed under CC BY 4.0.

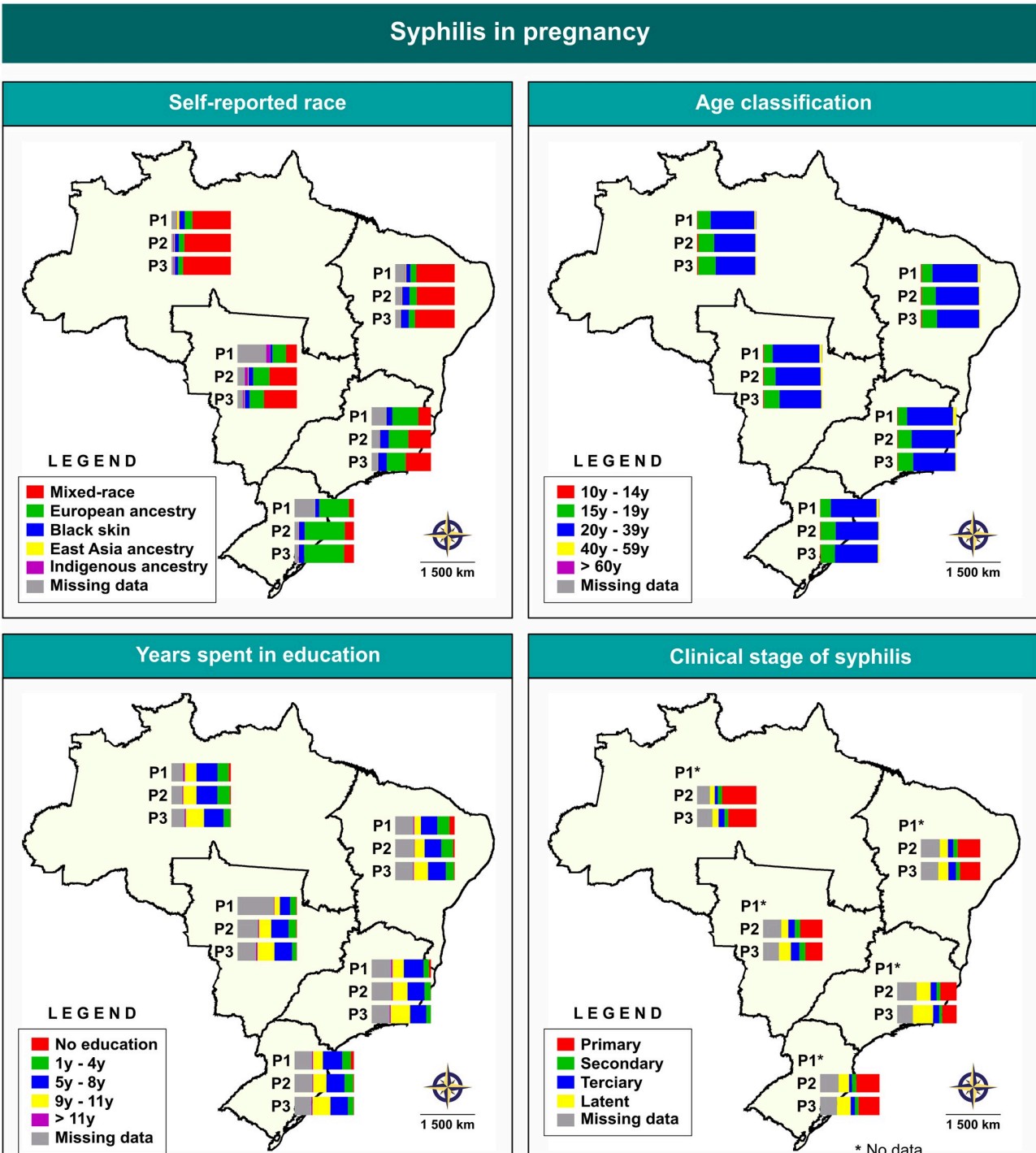

**Fig 4. Analysis of changes in sociodemographic profile by region, stratified according to three periods of SiP notifications.** P1*: data not available. Digital maps in the public domain (publicly available) were obtained IBGE cartographic database in shapefile format (.shp), which was subsequently reformatted and analyzed using QGIS version 3.10. Authors specify that this figure is licensed under CC BY 4.0.

cases were concentrated in pregnant women who had up to eight years of formal schooling in all three periods. With respect to clinical stage of syphilis, in P2 most pregnant women had primary syphilis (over 40% in all macroregions), while in P3 primary syphilis was predominant in the North (47.6%), Northeast (34.5%), South (35.2%) and Central-West (29%). Most cases of latent syphilis were concentrated in the Southeast (34.1%) in P3. No data was available in the SINAN database during period 1.

## Spatiotemporal distribution analysis

Fig 5 (S3 Table) depicts the spatiotemporal distribution of reported numbers of CS cases at 15 distinct time points. Overall, significant changes in the incidence throughout Brazil are evidenced over time. Between 2002 and 2016, infection rates increased significantly from 143 to 891 per 100,000 live births/year. The number of positive cases was found to increase in some microregions in Espírito Santo, Sergipe, Pernambuco, Amapá, and Tocantins states, especially until 2009. The high numbers of cases were reported in the microregions of Linhares-ES (9,119 cases per 100,000 live births), Contiguiba-SE (7,952 cases per 100,000 live births) and Recife-PE (7,612 cases per 100,000 live births). Interestingly, from 2009, increasing numbers of cases were notified in Recife-PE (15,801 cases per 100,000 live births), Natal-RN (15,027 cases per 100,000 live births), and Botucatu-SP (14,703 cases per 100,000 live births). However, from 2009, the cases of CS achieved a significant increase in almost all states, highlighting all five Brazilian macroregions.

The spatiotemporal distribution of reported rates of SiP is showed at 15 distinct time points in Fig 6 (S4 Table). SiP rates were low in the country, in 2002 and 2003 years. Between 2002 and 2016, infection rates increased significantly from 0.02 to 19.1 per 100,000 inhabitants/ year. In the North macroregion, Japurá-AM microregion shown high-incidence from 2005 to 2016 reaching a peak of 5,122 cases per 1000,000 inhabitants in 2016. The Cassilândia-MS microregion also presented an increased in the numbers reaching 671 cases per 1000,000 inhabitants in 2016. On the other hand, from 2012 the number of positive cases was found to increase in some microregions of at least 13 states, as Santo Antônio de Pádua-RJ (1,994 cases per 1000,000 inhabitants in 2016), Itaguara-MG (1,278 cases per 1000,000 inhabitants in 2016), Jeremoabo-BA (791 cases per 1000,000 inhabitants in 2016) and Ibaiti-PR (683 cases per 1000,000 inhabitants in 2016). From 2001 to 2018 the RR of SiP increased about 400% in the country (RR: 1,00 to 445,50), highlighting only 556 microregions, except Oiapoque-AP and Brasília-DF which presented zero rates.

## Spatial distribution of syphilis cases

Fig 7 (S5 Table) illustrates the total number of cumulative cases of syphilis at the end of the evaluated periods, with microregions used as units of analysis. Of the 558 microregions evaluated between 2001 and 2017, 549 (98.4%) and 558 (100%) reported at least one case of CS and SiP, respectively. The spatial distribution of CS indicated that the Recife-PE microregion had the highest cumulative incidence (2,075/100,000 live births). High numbers of cases were also reported in other microregions: Natal-RN (19,898/100,000 live births), Cotinguiba-SE (19,333/ 100,000 live births) and Rio de Janeiro-RJ (18,276/100,00 live births). With regard to SiP, higher incidences were observed in the Japurá-AM (20,208/100,000 inhabitants), Santo Antônio de Pádua-RJ (14,717/100,000 inhabitants) and Itaguara-MG (6,544/100,000 inhabitants) microregions.

## Discussion

We performed a systematic spatiotemporal analysis using secondary data of reported cases of CS and SiP in Brazil during the respectively studied period. Overall, our results indicate the

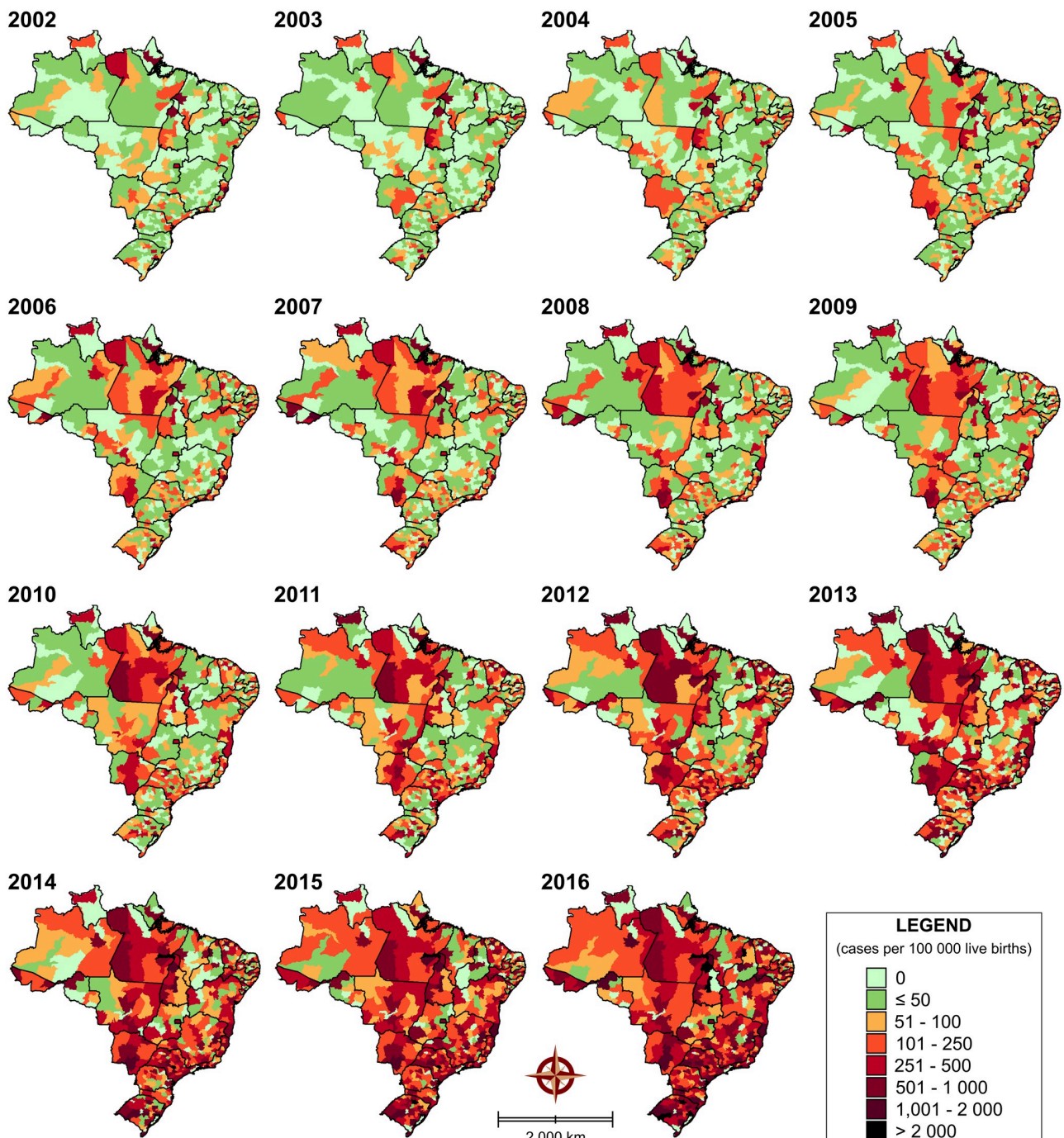

**Fig 5. Spatiotemporal distribution of CS cases (2002 to 2016) in Brazil, based on cases by microregions.** Digital maps in the public domain (publicly available) were obtained from IBGE cartographic database in shapefile format (.shp), which was subsequently reformatted and analyzed using QGIS version 3.10. Authors specify that this figure is licensed under CC BY 4.0.

existence of an increasing in case numbers over time, that there was an upward trend in the number of cases de CS e SiP (2011 to 2017), which corroborates the results presented here [20, 21]. However, in fact the number of cases may be either a reflection of increasing of cases notification or of infected people. Regardless, the epidemy witnessed in recent years suggests that

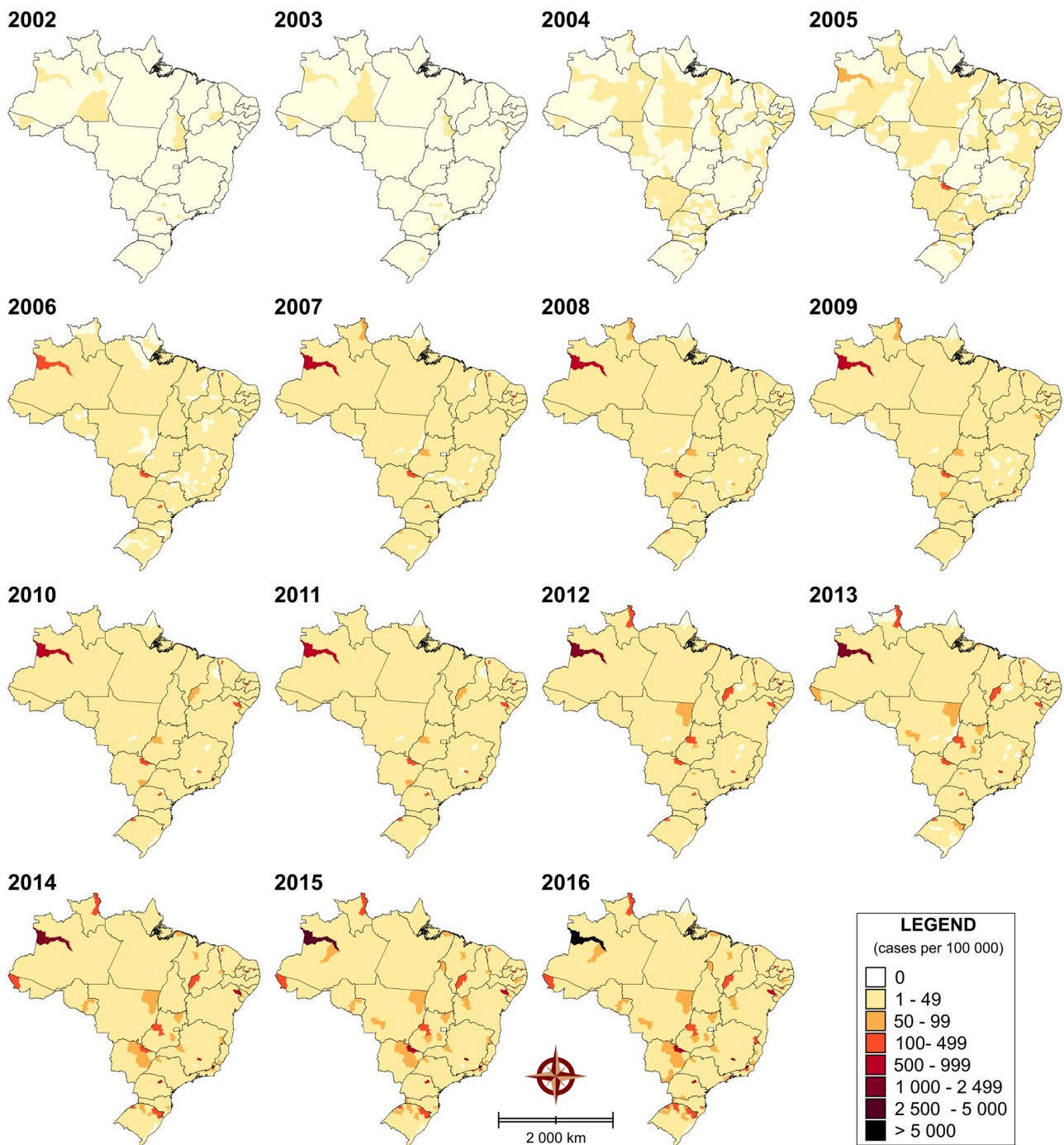

**Fig 6. Spatiotemporal distribution of SiP cases (2001 to 2017) in Brazil, based on cases by microregions.** Digital maps in the public domain (publicly available) were obtained from IBGE cartographic database in shapefile format (.shp), which was subsequently reformatted and analyzed using QGIS version 3.10. Authors specify that this figure is licensed under CC BY 4.0.

syphilis should be "prioritized" as a national public health concern due to the dramatic increases in rates of SiP and CS.

Between 2001 and 2017, 235,895 SiP notifications (6.33 cases/100,000 inhabitants) were reported, followed by 188,630 notifications for CS (359 cases/100,000 live births). These data

## Congenital syphilis

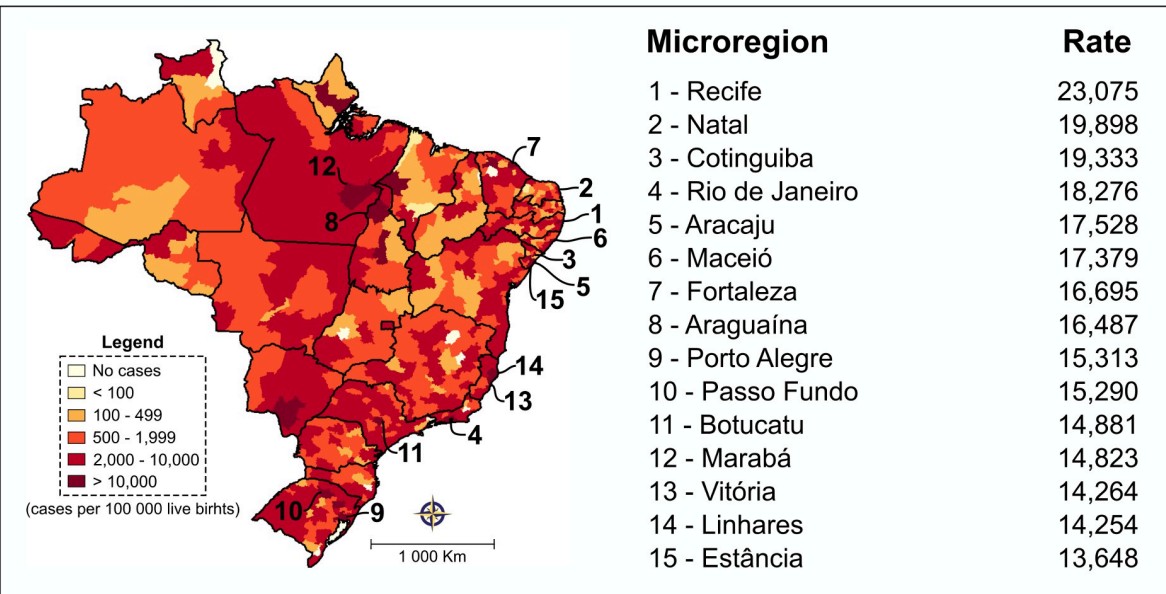

| Microregion | Rate |
|---|---|
| 1 - Recife | 23,075 |
| 2 - Natal | 19,898 |
| 3 - Cotinguiba | 19,333 |
| 4 - Rio de Janeiro | 18,276 |
| 5 - Aracaju | 17,528 |
| 6 - Maceió | 17,379 |
| 7 - Fortaleza | 16,695 |
| 8 - Araguaína | 16,487 |
| 9 - Porto Alegre | 15,313 |
| 10 - Passo Fundo | 15,290 |
| 11 - Botucatu | 14,881 |
| 12 - Marabá | 14,823 |
| 13 - Vitória | 14,264 |
| 14 - Linhares | 14,254 |
| 15 - Estância | 13,648 |

## Syphilis in pregnancy

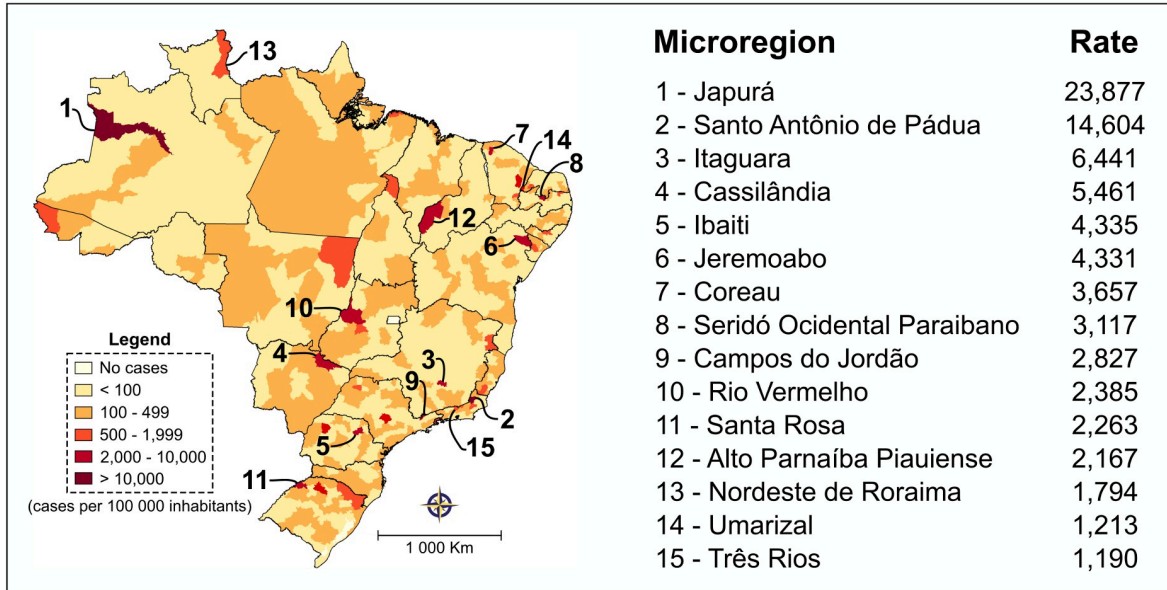

| Microregion | Rate |
|---|---|
| 1 - Japurá | 23,877 |
| 2 - Santo Antônio de Pádua | 14,604 |
| 3 - Itaguara | 6,441 |
| 4 - Cassilândia | 5,461 |
| 5 - Ibaiti | 4,335 |
| 6 - Jeremoabo | 4,331 |
| 7 - Coreau | 3,657 |
| 8 - Seridó Ocidental Paraibano | 3,117 |
| 9 - Campos do Jordão | 2,827 |
| 10 - Rio Vermelho | 2,385 |
| 11 - Santa Rosa | 2,263 |
| 12 - Alto Parnaíba Piauiense | 2,167 |
| 13 - Nordeste de Roraima | 1,794 |
| 14 - Umarizal | 1,213 |
| 15 - Três Rios | 1,190 |

**Fig 7. Spatial distribution of CS and SiP (2001 to 2017) in microregions.** The top 15 states or microregions with the highest cumulative rates per 100,00 live births for CS and 100,000 inhabitants SiP. Digital maps in the public domain (publicly available) were obtained from IBGE cartographic database in shapefile format (.shp) and reformatted and analyzed using QGIS version 3.10. Authors specify that this figure is licensed under CC BY 4.0.

are consistent with analyses published by the Ministry of Health (MoH), which indicate an increase from 1.7 to 8.6 cases/1,000 live births for CS between 2003 and 2017 and 0.5 to 17.2 cases per 1,000 live births for SiP between 2005 and 2017, with SiP only becoming reportable in 2005. According to a systematic review study carried out in Brazil (from 2002 to 2017), it is noted that the prevalence of SiP and CS in country remained above the 2016 target established by the World Health Organization (WHO) [22]. The strategy to combat STIs from 2016 to

2021 prioritizes the elimination of CS by implementing comprehensive syphilis screening and treatment among pregnant women, with a goal of reducing the global incidence of syphilis by 90%, with 50 or fewer cases of CS per 100,000 live births in 80% of the countries worldwide by 2030 [23]. Furthermore, in comparison to data published in international studies, the incidence rates of CS are significantly higher in Brazil than those reported in other countries [3].

It is important to note that changes in the epidemiologic infection profile in recent years are likely associated with a) increased testing coverage enabled by the adoption of rapid diagnostic tests, which therefore allowed for the expanded identification of incident cases throughout the country [24–26], b) the shortage of benzathine penicillin, which, since 2014, has affected Brazil, as well as other countries, due to a deficit of raw materials required for its production [27], and c) implementation of a governmental program in 2011 called "Rede Cegonha" (stork program), which expanded early detection efforts as a result of periodic screening for prenatal, intrapartum and postpartum syphilis [28]. Consequently, this shortage contributed to increases in untreated or inadequately treated SiP, which directly led to higher rates of CS [25]. Conversely, surveillance data also show that the general profiles of those infected contain reports of high numbers of sexual partners, unprotected sex, recreational drug use and the use of sex-oriented social networking [29, 30].

The present study also investigated the epidemiological profile of syphilis in both maternal and newborn populations in Brazil. A similar prevalence (~ 50%) was also observed between both male and female cases of CS in the south of Brazil, which found no significant differences in incidence according to sex. This can be explained by the fact that CS is transmitted vertically and that infection transmissibility is influenced by the mother's infection stage and fetal exposure [31]. The majority of reported CS cases occurred among newborns identified by the family as 'mixed-race' similar to other reports [31–34].

Herein, most cases of CS were diagnosed in asymptomatic children less than seven days after birth with 70% mothers reported receiving prenatal care. However, the high coverage of hospital births with a notification system for CS based on maternity services are the main reason why diagnosis usually occurs within seven days after birth (typically between the 1st and 2nd day of life) [33, 34]. Interestingly, it was found that despite the predominance of prenatal care in 70+% of the CS cases investigated herein, significantly high numbers of cases of CS were nonetheless reported throughout the country. While prenatal care is becoming more commonplace in Brazil, this rate still remains below the recommendations established by the MoH, which advocate that prenatal care must be properly provided to all pregnant women [35]. It is known that the incidence of syphilis is an important indicator of accessibility and prenatal care quality [28].

Despite the expansion of diagnosis and treatment in Brazil, increases in the number of cases indicates shortcomings in the efforts designed to control and prevent this STI [36]. At the same time, health authorities have also attributed increases in incidence to the success of public health actions in improving detection rates. Nevertheless, actions designed to improve health care access for pregnant women have performed poorly in terms of CS prevention [25]. Early diagnosis and treatment of SiP can reduce CS-related cases, such as miscarriages, stillbirths, and infant deaths [37]. Therefore, it is important that all pregnant women be tested at the first prenatal visit scheduled (1st trimester), with 28 weeks pregnant and at the time of delivery to promptly implement appropriate therapy if necessary [32, 38–42]. Information regarding CS vertical transmission should be provided to pregnant women at the onset of prenatal care, and physicians must inform patients regarding the risks and consequences of the disease to the mother and her fetus [32, 42]. Nonetheless, difficulties in diagnosis, interruption of and/or low attendance in prenatal examinations, non-adherence or inadequate treatment

regarding antibiotic dosage, and absence of partners at prenatal appointments have been reported as risk factors for the development and non-interruption of syphilis [42–47].

Regarding the sociodemographic profile of pregnant women observed herein, most cases were identified among women aged 20–39 years who self-identified as 'mixed-race', had up to eight years of formal education and were mainly diagnosed with primary syphilis. However, it worth noting that many records contained missing self-reported skin color classification data in P1, which can be considered as a bias in the interpretation of our results. Indeed, the data presented herein corroborate other studies [25, 31–34, 36, 37, 43–46, 48–53] that identified a significant correlation between SiP and women who dropped out of school, self-identified as 'black' or 'mixed-race', age below 20 years or between 20 and 30 years old [48], without access to quality health services, preventive and educational programs or received assistance at public health care units, including adequate prenatal care [34, 44, 45, 54, 55]. As for cases of primary syphilis in pregnant women, it is noteworthy that like the misclassification of latent syphilis in the United States [56], the estimated number of 40% found here may also have been misclassified, as this is considered a major problem. in Brazil. In fact, the correct classification of the clinical stage of the infection depends on experienced medical personnel.

The association between teenagers (low-income and with primary and/or secondary syphilis) and cases of CS strengthens the hypothesis this biological and sociodemographic characteristic associated with less prenatal care in young women can be a better explanation for higher rates of CS in this population [50, 52]. A low education level is also considered a marker of greater risk exposure due to unawareness regarding the importance of prevention. In some settings in Brazil, higher frequencies of syphilis were diagnosed during the 2$^{nd}$ and 3$^{rd}$ trimesters of pregnancy, possibly related to (I) delayed initiation of prenatal care and (II) substandard quality of obstetric care [51]. These observations reaffirm the importance of early syphilis detection in pregnant women, as well as the availability of appropriate treatment women and their partners [51].

The present study identified spatial clusters of municipalities with high rates of CS and SiP in Brazil. Between 2001 and 2017, almost all regions and microregions of the country reported a higher intensity of CS and SiP infections [47, 54, 57–62]. However, in most of Brazil's microregions, there are few studies of these regions that do not highlight these areas as priority for a syphilis surveillance study. Considering the epidemic profile of Brazil, we highlight some priority areas in which intervention, such as appropriate patient management and effective infection control measures, could prove beneficial. The observed variations in incidence among the municipalities may be the result of a decline in the underreporting of cases or reflect problems in local health systems, such as a lack of access to specialized services. Importantly, incomplete reporting hinders the elaboration of preventive strategies by policymakers, resulting in ineffective epidemiological surveillance [53, 63]. It is evident that the Brazilian healthcare system will continue to be challenged by this scenario, as despite government investment in awareness campaigns, the circumstances remain far from ideal.

We suggest that distinct strategies are required to reach more vulnerable populations and to minimize inequalities that enable greater access to health services. Poverty prompts specific vulnerabilities, whether behavioral or brought on by deficiencies in health services, such as prenatal care access and quality, which are also significantly associated with SiP. Brazilian social inequality in health supports the hypothesis that the prevalence of SiP is associated with a lower socioeconomic status [34, 44, 45]. SiP control programs should place greater focus on these more vulnerable populations [44, 45], especially considering that the lack of or inadequacies in public sexual education policies for younger individuals was associated with decreased condoms use in casual sexual relations in recent years [34].

As a limitation, we consider that by syphilis is a reemerging infection with epidemic behavior in Brazil, the numbers of reported cases vary widely, indicating possible underreporting. This problem affects the results and interpretation of epidemiological studies, since it is not possible to draw concrete conclusions about the effectiveness of diagnosis and treatment of patients in the public health network, and about the implementation and correct monitoring of prenatal care. Mandatory reporting of cases helps in the evaluation of public health policies. On the other hand, both acquired syphilis, including in pregnant women, and congenital syphilis pose the same problem.

In this way, concluded despite the existence of control and awareness programs for STIs, current measures have proven ineffective in decreasing the incidence of CS and SiP in Brazil. Additionally, the underreporting of registered cases throughout the country results in biased data analysis. Considering the epidemiological profile of infection in Brazil, our results highlight the need to enhance preventive and control measures for eradication of syphilis.

## Supporting information

**S1 Checklist. STROBE checklist.**
(DOCX)

**S1 Table. Sociodemographic variables of CS cases.**
(XLSX)

**S2 Table. Sociodemographic variables of SiP cases.**
(XLSX)

**S3 Table. Spatiotemporal distribution of reported numbers of CS cases.**
(XLSX)

**S4 Table. Spatiotemporal distribution of reported numbers of SiP cases.**
(XLSX)

**S5 Table. Spatial distribution of CS and SiP cases.**
(XLSX)

## Acknowledgments

We express our gratitude to Andris K. Walter for providing English language revision and manuscript copyediting assistance.

## Author Contributions

**Conceptualization:** Fred Luciano Neves Santos.

**Data curation:** Fred Luciano Neves Santos.

**Formal analysis:** Ângelo Antônio Oliveira Silva, Carlos Gustavo Regis-Silva, Fred Luciano Neves Santos.

**Funding acquisition:** Fred Luciano Neves Santos.

**Investigation:** Ângelo Antônio Oliveira Silva, Wayner Vieira de Souza, Emily Ferreira Santos, Carlos Gustavo Regis-Silva, Fred Luciano Neves Santos.

**Methodology:** Ângelo Antônio Oliveira Silva, Wayner Vieira de Souza, Larissa de Carvalho Medrado Vasconcelos, Fred Luciano Neves Santos.

**Project administration:** Fred Luciano Neves Santos.

**Resources:** Fred Luciano Neves Santos.

**Supervision:** Fred Luciano Neves Santos.

**Validation:** Ângelo Antônio Oliveira Silva, Carlos Gustavo Regis-Silva, Fred Luciano Neves Santos.

**Visualization:** Ramona Tavares Daltro, Fred Luciano Neves Santos.

**Writing – original draft:** Ângelo Antônio Oliveira Silva, Leonardo Maia Leony, Natália Erdens Maron Freitas, Ramona Tavares Daltro, Emily Ferreira Santos, Larissa de Carvalho Medrado Vasconcelos, Carlos Gustavo Regis-Silva, Fred Luciano Neves Santos.

**Writing – review & editing:** Ângelo Antônio Oliveira Silva, Leonardo Maia Leony, Wayner Vieira de Souza, Natália Erdens Maron Freitas, Maria Fernanda Rios Grassi, Carlos Gustavo Regis-Silva, Fred Luciano Neves Santos.

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
