## [Decision Letter · Decision Letter 0]

28 Feb 2022

PONE-D-21-31051

Spatiotemporal distribution analysis of syphilis in Brazil: Cases of congenital and syphilis in pregnant women from 2001-2017

PLOS ONE

Dear Dr. Santos,

Thank you for submitting your manuscript to PLOS ONE. After careful consideration, we feel that it has merit but does not fully meet PLOS ONE’s publication criteria as it currently stands. Therefore, we invite you to submit a revised version of the manuscript that addresses the points raised during the review process.

We look forward to receiving your revised manuscript.

Kind regards,

Antonio Simone Laganà, M.D., Ph.D.

Academic Editor

PLOS ONE

https://journals.plos.org/plosone/s/fileid=ba62/PLOSOne_formatting_sample_title_authors_affiliations.pdf".

“The authors declare that they have no competing interests.”

4. We note that you have included the phrase “data not available” in your manuscript. Unfortunately, this does not meet our data sharing requirements. PLOS does not permit references to inaccessible data. We require that authors provide all relevant data within the paper, Supporting Information files, or in an acceptable, public repository. Please add a citation to support this phrase or upload the data that corresponds with these findings to a stable repository (such as Figshare or Dryad) and provide and URLs, DOIs, or accession numbers that may be used to access these data. Or, if the data are not a core part of the research being presented in your study, we ask that you remove the phrase that refers to these data.

5. We note that [Figures 1, 3-7] in your submission contain [map/satellite] images which may be copyrighted. All PLOS content is published under the Creative Commons Attribution License (CC BY 4.0), which means that the manuscript, images, and Supporting Information files will be freely available online, and any third party is permitted to access, download, copy, distribute, and use these materials in any way, even commercially, with proper attribution. For these reasons, we cannot publish previously copyrighted maps or satellite images created using proprietary data, such as Google software (Google Maps, Street View, and Earth). For more information, see our copyright guidelines: http://journals.plos.org/plosone/s/licenses-and-copyright.

a. You may seek permission from the original copyright holder of [Figures 1, 3-7] to publish the content specifically under the CC BY 4.0 license. 

Additional Editor Comments:

The topic of the manuscript is interesting. Nevertheless, the reviewers raised several concerns: considering this point, I invite authors to perform the required major revisions.

Reviewers' comments:

Reviewer's Responses to Questions

**Comments to the Author**

1. Is the manuscript technically sound, and do the data support the conclusions?

Reviewer #1: Partly

Reviewer #2: Yes

Reviewer #3: Yes

2. Has the statistical analysis been performed appropriately and rigorously? 

Reviewer #1: Yes

Reviewer #2: Yes

Reviewer #3: Yes

3. Have the authors made all data underlying the findings in their manuscript fully available?

Reviewer #1: Yes

Reviewer #2: Yes

Reviewer #3: Yes

4. Is the manuscript presented in an intelligible fashion and written in standard English?

Reviewer #1: Yes

Reviewer #2: Yes

Reviewer #3: Yes

5. Review Comments to the Author

Reviewer #1: The article presents a relevant topic using novel data with sub-national and temporal detail, potentially advancing the current understanding of CI and SiP’s geographical diffusion and temporal evolution. I have read it with great interest and enjoyed the manuscript in general. However, there is room for improvement in various parts, in particular: the literature review should be extended, the data section needs to be more detailed, the methods should include spatial econometric techniques, and the results and discussion sections should better elaborate on the implications of the study results and their impact on current research. (see document attached for further details)

Reviewer #2: This analysis and findings are valuable for the surveillance of maternal and congenital syphilis Brazil. These findings are of considerable concern regarding the risk of syphilis in women and unborn infants.

There are many opportunities to shorten and streamline the discussion and provide updated references to reflect national, regional and global efforts to eliminate mother to child transmission of syphilis

Please find attached my comments that exceed the character count.

Reviewer #3: The manuscript PONE-D-21-31051, entitled "Spatiotemporal distribution analysis of syphilis in Brazil: Cases of congenital and syphilis in pregnant women from 2001-2017" is an interesting paper and highlights an important problem for public health in Brazil. I have few comments and suggestions:

a) Pag. 3, lines 56-58: Brazilian data in the first paragraph should be updated. The last data are from 2020.

b) Pag.3, lines 69-70: Even if the paper is about syphilis in pregnancy and congenital syphilis, it should be included that there is an ordinance for syphilis in adults (acquired syphilis). It is an infection of compulsory notification since 2010.

c) Pag. 5, line 115: Change “Brazilian National Census” to Brazilian National Estimates”. The last census was in 2010.

d) Page 9, lines 220-221: 40% of the clinical stage of syphilis were classified as primary. The misclassification is a big problem in Brazil and the authors should emphasize it in the discussion section.

e) Page 11, lines 252-256: The authors describe data of syphilis in pregnancy during pregnancy. In page 13, lines 300-308, the authors need to highlight in the discussion that the compulsory notification started in 2005 for pregnant women.

f) Page 15, lines 350-355: Are there links to access these documents listed in Portuguese? If not, they could be excluded.

g) Page 15, lines 365-368: Authors explain about syphilis tests during pregnancy. It is important to include the recommendation to test for syphilis during labor. Both WHO and Brazilian guidelines recommend it.

h) The discussion section is a little to long, it should be optimized. There is discussion on demographics, as age, mixed-race, education, in three different parts.

i) I did not see in the discussion section a discussion about the use of secondary data. It is important to report it.

j) References should be revisited because they should follow the journal recommendations. There are different presentation of the number of authors and one of them are in caps letter.

6. PLOS authors have the option to publish the peer review history of their article (what does this mean?). If published, this will include your full peer review and any attached files.

Reviewer #1: No

Reviewer #2: No

Reviewer #3: No

---

## [Author Response · Author response to Decision Letter 0]

13 Jul 2022

Salvador-Bahia, Brazil - April 2022

Dear Dr. Laganà,

Editor Plos One

We thank you and the reviewers again for your thoughtful suggestions that helped improve our manuscript. We believe that this revised manuscript is a better and more balanced representation of our research, and we hope that it is now suitable for publication in your journal. The answers to the questions can be found below (PONE-D-21-31051).

Sincerely,

Dr. Fred Luciano Neves Santos

corresponding author

fred.santos@fiocruz.br

Author's Reply to the Review Report (Reviewer 1)

The article presents a relevant topic using novel data with sub-national and temporal detail, potentially advancing the current understanding of CI and SiP’s geographical diffusion and temporal evolution. I have read it with great interest and enjoyed the manuscript in general. However, there is room for improvement in various parts, in particular: the literature review should be extended, the data section needs to be more detailed, the methods should include spatial econometric techniques, and the results and discussion sections should better elaborate on the implications of the study results and their impact on current research. (see document attached for further details)

Question 1. The references need to be formatted properly (a lot of the journal names are shortened),

and often the article or volume number is missing. E.g. reference number 35 should be: Domingues RM, Szwarcwald CL, Souza PR, Leal MD. Prevalence of syphilis in pregnancy and prenatal syphilis testing in Brazil: birth in Brazil study. Revista de saúde pública. 2014;48(5): 766-74. In one occasion you cite WHO and others World Health Organization, it should be consistent throughout the references. Reference number 10 has the authors’ names listed in capital letters.

Reply: We thank the reviewer for bringing the standardization of references to our attention. We have formatted the references according to the instructions at https://www.frontiersin.org/about/author-guidelines using the Harvard Reference Style (author-date).

Question 2. The literature is missing a few important publications, their careful reading and inclusion will definitely improve the manuscript. See for instance, but not exclusively Marques dos Santos, M., Lopes, A. K. B., Roncalli, A. G., & Lima, K. C. D. (2020). Trends of syphilis in Brazil: a growth portrait of the treponemic epidemic. Plos one, 15(4), e0231029. Rgjh; Soares, K. K. S., Prado, T. N. D., Zandonade, E., Moreira-Silva, S. F., & Miranda, A. E. (2020). Spatial analysis of syphilis in pregnancy and congenital syphilis in the state of Espírito Santo, Brazil, 2011-2018. Epidemiologia e Serviços de Saúde, 29.

Reply: Thanks for suggesting new references. More recent studies were included in the text.

Before: “Overall, our results indicate the existence of a distinct trend for each form of syphilis, with an increasing tendency in case numbers over time [...] The high number of cases of CS and SiP observed from 2011 to 2017 could have also been due to the implementation of a governmental program in 2011 called "Rede Cegonha" (stork program), which expanded early detection efforts as a result of periodic screening for prenatal, intrapartum and postpartum syphilis [16].”

After (line 315-318): Overall, our results indicate the existence of an increasing in case numbers over time. that there was an upward trend in the number of cases de CS e SiP (2011 to 2017), which corroborates the results presented here [20,21].

Reference # 20 (line 503-505): Dos Santos MM, Lopes AKB, Roncalli AG, De Lima KC. Trends of syphilis in Brazil: A growth portrait of the treponemic epidemic. PLoS One. 2020;15: 1–11. doi:10.1371/journal.pone.0231029.

Reference # 21 (line 506-509): Lino CM, da Luz Rosário de Sousa M, Batista MJ. Epidemiological profile, spatial distribution, and syphilis time series: A cross-sectional study in a Brazilian municipality. J Infect Dev Ctries. 2021;15: 1462–1470. doi:10.3855/jidc.13780.

Before: “It is important to note that changes in the epidemiologic infection profile in recent years are likely associated with a) increased testing coverage enabled by the adoption of rapid diagnostic tests, which therefore allowed for the expanded identification of incident cases throughout the country […] [19]’’.

After (line 336-339): “It is important to note that changes in the epidemiologic infection profile in recent years are likely associated with a) increased testing coverage enabled by the adoption of rapid diagnostic tests, which therefore allowed for the expanded identification of incident cases throughout the country [24–26] […]”

Reference # 24 (line 516-519): Roncalli AG, Rosendo TMS de S, Santos MM Dos, Lopes AKB, Lima KC de. Effect of the coverage of rapid tests for syphilis in primary care on the syphilis in pregnancy in Brazil. Rev Saude Publica. 2021;55: 94. doi:10.11606/s1518-8787.2021055003264.

Reference # 26 (line 524-527): De Figueiredo DCMM, De Figueiredo AM, De Souza TKB, Tavares G, De Toledo Vianna RP. Relationship between the supply of syphilis diagnosis and treatment in primary care and incidence of gestational and congenital syphilis. Cad Saude Publica. 2020;36: e00074519. doi:10.1590/0102-311X00074519.

Before: “An established diagnosis does not guarantee adhesion to appropriate treatment by the patient. The late onset of symptoms (mostly detected in the third trimester), the interruption of and/or low attendance in prenatal examinations, difficulties in diagnosis, a lack of information regarding infection and unsafe sexual practices have been reported as risk factors for the development of syphilis [34–36]. Furthermore, errors in antibiotic dosage have also been of concern. Other key factors, such as conjugal infidelity, the absence of partners at prenatal appointments and reluctance in adhering to treatment protocols have also been reported by patients [26,33].”.

After (line 380-383): “Nonetheless, difficulties in diagnosis, interruption of and/or low attendance in prenatal examinations, non-adherence or inadequate treatment regarding antibiotic dosage, and absence of partners at prenatal appointments have been reported as risk factors for the development and non-interruption of syphilis [42–47].”

Reference # 47 (line 596-599): Amorim EKR, Matozinhos FP, Araújo LA, Silva TPR da. Tendência dos casos de sífilis gestacional e congênita em Minas Gerais, 2009-2019: um estudo ecológico. Epidemiol e Serv saude Rev do Sist Unico Saude do Bras. 2021;30: e2021128. doi:10.1590/S1679-49742021000400006.

Before: “In fact, STI and SiP were found to be more strongly associated with women who dropped out of school, self-identified as 'black' or 'mixed-race', were under 20 years of age or between 20 and 30 years old [41], had limited access to quality health services, preventive and educational programs or received assistance at public health care units, but without adequate prenatal care [25,35,36].”

After (line 390-394): “[…] that identified a significant correlation between SiP and women who dropped out of school, self-identified as 'black' or 'mixed-race', age below 20 years or between 20 and 30 years old [48], without access to quality health services, preventive and educational programs or received assistance at public health care units, including adequate prenatal care [34,44,45,54,55].”

Reference # 54 (line 624-626): Bezerra MLDMB, Fernandes FECV, Nunes JPDO, Baltar SLSMDA, Randau KP. Congenital syphilis as a measure of maternal and child healthcare, brazil. Emerg Infect Dis. 2019;25: 1469–1476. doi:10.3201/eid2508.180298.

Before: “The present study identified spatial clusters of municipalities with high rates of CS and SiP in Brazil. Between 2001 and 2017, almost all microregions of the country reported a higher intensity of CS and SiP infections.”

After (line 409-411): “The present study identified spatial clusters of municipalities with high rates of CS and SiP in Brazil. Between 2001 and 2017, almost all regions and microregions of the country reported a higher intensity of CS and SiP infections [47,54,57–62].”

Reference # 47 (line 596-599): Amorim EKR, Matozinhos FP, Araújo LA, Silva TPR da. Tendência dos casos de sífilis gestacional e congênita em Minas Gerais, 2009-2019: um estudo ecológico. Epidemiol e Serv saude Rev do Sist Unico Saude do Bras. 2021;30: e2021128. doi:10.1590/S1679-49742021000400006.

Reference # 54 (line 624-626): Bezerra MLDMB, Fernandes FECV, Nunes JPDO, Baltar SLSMDA, Randau KP. Congenital syphilis as a measure of maternal and child healthcare, brazil. Emerg Infect Dis. 2019;25: 1469–1476. doi:10.3201/eid2508.180298.

Reference # 57 (line 633-636): Medeiros JAR, Yamamura M, da Silva ZP, Domingues CSB, Waldman EA, Chiaravalloti-Neto F. Spatiotemporal dynamics of syphilis in pregnant women and congenital syphilis in the state of São Paulo, Brazil. Sci Rep. 2022;12. doi:10.1038/s41598-021-04530-y.

Reference # 58 (line 637-641): Nunes PS, Guimarães RA, Rosado LEP, Marinho TA, Aquino ÉC de, Turchi MD. Tendência temporal e distribuição espacial da sífilis gestacional e congênita em Goiás, 2007-2017: um estudo ecológico. Epidemiol e Serv saude Rev do Sist Unico Saude do Bras. 2021;30: e2019371. doi:10.1590/S1679-49742021000100002.

Reference # 59 (line 642-645): Soares MAS, Aquino R. Completude e caracterização dos registros de sífilis gestacional e congênita na Bahia, 2007-2017. Epidemiol e Serv saude Rev do Sist Unico Saude do Bras. 2021;30: e20201148. doi:10.1590/S1679-49742021000400018.

Reference # 60 (line 646-649): De Souza TA, Teixeira KK, Santana RL, Penha CB, Medeiros ADA, De Lima KC, et al. Intra-urban differentials of congenital and acquired syphilis and syphilis in pregnant women in an urban area in northeastern Brazil. Trans R Soc Trop Med Hyg. 2021;115: 1010–1018. doi:10.1093/trstmh/trab011

Reference # 61 (line 650-654): de Mélo KC, Dos Santos AGG, Brito AB, de Aquino SHS, Alencar ÉTDS, Duarte EM da S, et al. Syphilis among pregnant women in Northeast Brazil from 2008 to 2015: A trend analysis according to sociodemographic and clinical characteristics. Rev Soc Bras Med Trop. 2020;53: 1–6. doi:10.1590/0037-8682-0199-2019.

Reference # 62 (line 655-658): Soares KKS, Prado TN do, Zandonade E, Moreira-Silva SF, Miranda AE. Spatial analysis of syphilis in pregnancy and congenital syphilis in the state of Espírito Santo, Brazil, 2011-2018. Epidemiol e Serv saude Rev do Sist Unico Saude do Bras. 2019;28: e2018197. doi:10.5123/S1679-49742019000300005.

Question 3. The paper does not present any reference to the spatial nature of the data from an econometric point of view. It would be useful to add Global and Local Moran’s I (or similar indexes) to measure the non randomness of spatial clusters, especially using LISA plots and bivariate LISA plots. See for instance: Brooks, M. M. (2019). The advantages of comparative LISA techniques in spatial inequality research: Evidence from poverty change in the United States. Spatial Demography, 7(2), 167-193.; Yin, F., Feng, Z., & Li, X. (2012). Spatial analysis of primary and secondary syphilis incidence in China, 2004-2010. International journal of STD & AIDS, 23(12), 870-875, Graepp Fontoura, I., Lima, V. C., Fontoura, V. M., Santos, F. S., de Jesus Costa, A. C. P., de Oliveira, F. J. F., ... & Santos Neto, M. (2021). Spatial analysis of congenital syphilis in a federative unit in northeastern Brazil. Transactions of The Royal Society of Tropical Medicine and Hygiene, 115(10), 1207-1217. And more in general: Anselin, L. (2001). Spatial econometrics. A companion to theoretical econometrics, 310330.

Reply: We thank the reviewer for his/her criticism and politely ask him/her to reconsider this opinion. The implicit hypothesis of the calculation of the Moran index is the stationary of the first and second order, and the index loses its validity when calculated for nonstationary data. When there is a non-stationary of first order (trend), the neighbors will tend to have closer values than the ones more distant because each value is compared to the global average, inflating the index. For more details, please read the paper below:

Naizer CCBR, Rodrigues DS, Pedreira Junior JU, Pitombo CS. G-SIVAR: a global spatial indicator based on variogram. Bol. Ciênc. Geod. 25 (4), 2019.

https://doi.org/10.1590/s1982-21702019000400022

Question 4. In the abstract, you mention that CS and SiP is more prevalent among mixed race individuals (newborns and women) but on p.8 the picture is different, and it is described to vary according to macro-regions. Different regions in Brazil have different ethnic compositions, you should make that clear earlier on.

Reply: We thank the reviewer for this comment. The following sentence has been revised for clarification:

Before: “The epidemiological profile of Brazil indicates most reported CS cases occurred among ‘mixed-race’ newborns who were diagnosed within seven days after birth and whose mothers had received prenatal care.”

After (line 37-40): In general, the epidemiologic profile of Brazil indicates most reported CS cases occurred in "mixed-race" newborns who were diagnosed within seven days of birth and whose mothers had received prenatal care, but the epidemiologic profile varies by Brazilian macroregion.

Question 5. Moreover, in the legend in Figures 3 and 4, you report the classifications as translated from Portuguese, which sounds off in English. The authors might want to revise and update that. I suppose 'Brancos' would be individuals of European descent, 'Amarelos' would be of East Asian descent, etc.

Reply: We have revised the shelf-reposted skin color throughout the manuscript and in the legends of Figures 3 and 4. We also added a sentence at the end of the Study Design subsection that reads as follows:

After: no text.

Before (line 147-150): Self-reported skin color was classified as European ancestry, dark skin, East Asian ancestry, indigenous ancestry, mixed-race (persons whose skin color is not classified as dark-skinned, European, indigenous, or East Asian).

Question 6. The Study design section would benefit from a more thorough description of the ethnic categories used. Mixed race is for any mixed race, as in anyone who does not identify as either of black, white, indigenous, or East Asian descent?

Reply: Yes, mixed-race refers to anyone that no shelf-reported skin color as dark skin, European ancestry, indigenous, or East Asian ancestry. In order to clarify the concept of “mixed-race”, we have added the following sentence in the end of the Study Design subsection:

After: no text.

Before (line 147-150): Self-reported skin color was classified as European ancestry, dark skin, East Asian ancestry, indigenous ancestry, mixed-race (persons whose skin color is not classified as dark-skinned, European, indigenous, or East Asian).

Question 7. In fig 3 and 4 you present bar charts over Brazil’s macro regions map. However, a lot of information is missed because of charts resizing to fit within each macro-region. I would arrange the bar charts without the map, and add for each macro-region the population composition (the main group). That way, it’d be easier to interpret CI and SiP prevalence in said category, whether it is just representative of the population composition or of a disadvantaged population group (I believe the South is predominantly white so it’s not a shocker that most CI children are white). One example that comes to mind is the share of Covid deaths among Alaskans, where 30% of Covid19-related deaths were registered among Alaskan Natives, although they represent 15% of the total Alaska population.

(https://coronavirus.jhu.edu/data/disparity-explorer). In figure 4: Years spent in education.

Reply: We thank the reviewer for this excellent suggestion. However, we would like to ask the reviewer to reconsider his/her criticism. The data available in the databases do not allow us to extract the national population (or each Brazilian region) according to the variables studied. Therefore, we would like to continue presenting the data as they were presented in the first version of the manuscript.

Question 8. In figure 4: Years spent in education.

Reply: We thank the reviewer for this comment. We have replaced "Years of study" with "Years spent in education".

Question 9. In the spatiotemporal description of results the regions mentioned may not be familiar with audiences outside of Brazil. Are those mainly rural, urban, tropical or large cities?

Reply: The names of the states in which the microregions are located are given in the text, e.g., SE, TO, AM, MG, etc. In addition, the 15 microregions with the most cases are shown on the map in Figure 7 so that the reader outside Brazil knows where they are located. We thank the reviewer for his/her criticism and politely ask him/her to reconsider this opinion. All microregions are tropical and consist of several rural and urban areas, with large and small cities.

Question 10. The results and discussion can be improved by contextualizing them. For instance, you mention Cassilandia, which is a small city at the border with Mato Grosso: elaborate on why its results are relevant when compared to other cities.

Reply: We thank the reviewer for this comment. Indeed, as mentioned, the results of Fig. 7 are sometimes not put into proper context in the discussion. These results show the main microregions in Brazil that, according to the Ministry of Health, had high incidence rates during the period studied. However, for most of these microregions, there are no published studies to support our discussion. Therefore, we discuss these results in general, as presented in lines 415-416. We may add one additional piece of information in this paragraph.

Before: “Between 2001 and 2017, almost all microregions of the country reported a higher intensity of CS and SiP infections.”

After (line 410-413): “Between 2001 and 2017, almost all regions and microregions of the country reported a higher intensity of CS and SiP infections [47,54,57–62]. However, in most of Brazil's microregions, there are few studies of these regions that do not highlight these areas as priority for a syphilis surveillance study.”

Question 11. How does under-reporting, and its decrease in the last 15 years affect the results and its interpretation? Are there population subgroups most likely to be missed?

Reply: Thank you for the relevant question. We have entered the answer to this as a paragraph of the discussion.

Before: No text.

After (line 433-440): “As a limitation, we consider that by syphilis is a reemerging infection with epidemic behavior in Brazil, the numbers of reported cases vary widely, indicating possible underreporting. This problem affects the results and interpretation of epidemiological studies, since it is not possible to draw concrete conclusions about the effectiveness of diagnosis and treatment of patients in the public health network, and about the implementation and correct monitoring of prenatal care. Mandatory reporting of cases helps in the evaluation of public health policies. On the other hand, both acquired syphilis, including in pregnant women, and congenital syphilis pose the same problem”.

Question 12. The article needs careful revision, there are typos throughout the text, as well as some

expressions that should be improved (see below a few examples).

Reply: We thank the reviewer for this comment. The English language was revised by Andris K. Walter, a native American speaker. The author's intent has not been distorted in any way by the revision.

Question 13. P3. line 60: condom-less sex (unprotected intercourse?)

Reply: Yes, condom-less sex means unprotected sex. In order to avoid any misinterpretation, we have changed the text as follows:

Before: “…is mainly transmitted through condomless sex (acquired syphilis) […]”.

After (line 66-67): ‘’…is mainly transmitted through unprotected intercourse (acquired syphilis) […]’’.

Question 14. P.3 line 62 space before full stop.

Reply: Thank you for bringing this error to our attention. We have removed the space before the period.

Before: “…by exposure to blood or contaminated body fluids [5–8]. In vertical …”

After (line 68): …by exposure to blood or contaminated body fluids [5–8]. In vertical …

Question 15. P.3 line 67 parenthesis misplaced.

Reply: Thank you for bringing this error to our attention. We have removed the incorrectly placed brackets.

Before: “Despite the existence of diagnostic tests and (effective antibiotic treatment…”

After (line 73): “Despite the existence of diagnostic tests and effective antibiotic treatment …’’

Question 16. Page 4 ends with a title.

Reply: We have fixed this formatting error. The title has been replaced at the top of the next page.

Question 17. Line 188: you don’t mention the male to female ratio for the general pop.

Reply: Line 188 refers to the annual percentage change in syphilis cases. The question about the ratio between males and females was not clear. We are available for further clarification.

Question 18. L. 201: suggesting a lack of protection against CS: lack of prenatal screening? Or

condom use?

Reply: We thank the reviewer for this comment. The expression "lack of protection against CS" refers to "lack of prenatal screening". We have adjusted the sentence accordingly.

Before: “With respect to prenatal care among the CS cases, over 70% reported receiving prenatal care regardless of period and macroregion, suggesting a lack of protection against CS.”

After: (line 223-225): “With respect to prenatal care among the CS cases, over 70% reported receiving prenatal care regardless of period and macroregion, lack of prenatal screening”.

Question 19 L. 211: you only mention here that race is self-reported, you should mention this in the data section as well

Reply: We have revised the shelf-reposted skin color throughout the manuscript and in the legends of Figures 3 and 4. We also added a sentence at the end of the Study Design subsection that reads as follows:

After: no text.

Before: (line 147-150): “Self-reported skin color was classified as European ancestry, dark skin, East Asian ancestry, indigenous ancestry, mixed-race (persons whose skin color is not classified as dark-skinned, European, indigenous, or East Asian)’’.

 

Author's Reply to the Review Report (Reviewer 2)

This analysis and findings are valuable for the surveillance of maternal and congenital syphilis Brazil. These findings are of considerable concern regarding the risk of syphilis in women and unborn infants. There are many opportunities to shorten and streamline the discussion and provide updated references to reflect national, regional and global efforts to eliminate mother to child transmission of syphilis. Please find attached my comments that exceed the character count.

Question 1. Major comment: Line 37 Abstract and in methods, results of paper. The WHO global estimates of CS use the number of live births as the denominator for the calculation of rate of congenital syphilis. In order to have these results considered at the regional and global levels, use of the global case rate definition would be of greater value. Also the WHO criteria for recognizing national elimination of mother to child transmission of syphilis (EMTCT) is <50 cases/100,000 ‘live births’. This analysis could contribute “directly” to Brazil’s national and provincial monitoring and surveillance of SiP and CS towards EMTCT if the rates of CS were calculated according to global case rate definitions.

a. https://www.who.int/publications/i/item/9789240039360

b. https://journals.plos.org/plosmedicine/article?id=10.1371/journal.pmed.1002329

c. https://www.who.int/reproductivehealth/congenital-syphilis/WHO-validation-EMTCT/en/

d. https://www.who.int/reproductivehealth/congenital-syphilis/emtc-gvac/en/

Reply: We thank the reviewer for this excellent proposal and fully agree with the suggestion to improve the presentation of the data. The data obtained from the new analyzes are included in the new version of the manuscript.

Question 2. Abstract line 45-46, please provide the values that increased 4000%... from what to what?

Reply: We thank the reviewer for this comment. We found an error in this information and made the correction suggested in the abstract (line 45-46) and in the results (line 279-280).

Abstract section:

Before: “… and the relative risk of SiP increased around 4,000%’’. 

After (line 46-47): … and the relative risk (RR) of SiP increased around 400% (RR: 1,00 to 445,50)’’

Results section:

Before: “From 2001 to 2018 the relative risk of SiP increased about 4,000% in the country…”

After (line 285-286): From 2001 to 2018 the RR of SiP increased about 400% in the country (RR: 1,00 to 445,50)…

Question 3. Introduction line 54, need to use the most recent WHO estimates for syphilis 

a. https://pubmed.ncbi.nlm.nih.gov/31384073/

b. Rowley J, Vander Hoorn S, Korenromp E, et al. Chlamydia, gonorrhoea, trichomoniasis and syphilis: global prevalence and incidence estimates, 2016 WHO Bulletin. 2019; 97:548-562.

Reply: Thank you for the suggestion to update the information. This data is now more current. We have made the suggested correction by changing the reference.

Before: “Increasing incidence and prevalence has been reported since 2008 in adults between 15 and 49 years of age, being 10.6 million cases of syphilis”.

After (line 54-57): According to the number of reported cases of curable STIs in 2016, only 6.3 million (1.67%) were syphilis in women and men aged 15-49 years. Looking at the period from 2012 to 2016, the estimated global prevalence was 0.5% and the incidence was estimated at 1.7 cases per 1,000 women and 1.6 cases per 1,000 men [2].

Question 4. Introduction line 56, need to use the most recent WHO estimates for congenital syphilis 

a. https://pubmed.ncbi.nlm.nih.gov/30811406/

Korenromp EL, Rowley J, Alonso M, et al. Global burden of maternal and congenital syphilis and associated adverse birth outcomes – estimates for 2016 and progress since 2012. PLOS One. 2019. 14(2): e0211720.

Reply: Thank you for your suggestion. We have made the suggested correction and changed the reference.

Before: “Around one million pregnant women become infected each year, resulting in approximately 300,000 fetal and neonatal deaths, placing more than 200,000 children at risk of premature death.”

After (line 57-63): In 2016, the estimated global prevalence of syphilis in pregnant women was 0.69%, resulting in a global congenital syphilis rate of 473 cases per 100,000 live births (~661,000 total cases). The data showed that maternal syphilis caused 143,000 early fetal deaths and stillbirths, 61,000 neonatal deaths, 41,000 preterm or low birth weight births, 109,000 infants with clinical congenital syphilis, and 306,000 cases of infants without clinical signs in mothers with untreated syphilis [3].

Question 5. Introduction: Line 63, would delete “following direct contact with bacterium by the mother”.

Reply: We thank the reviewer for his/her comment and have deleted the sentence as requested:

Before: “In vertical transmission from mother-to-child (CS), following direct contact with the bacterium by the mother, infection spreads to the fetus hematologically, predominantly via the transplacental route [9,10].”

After (line 68-70): In vertical transmission from mother-to-child (CS), the infection spreads to the fetus hematologically, predominantly via the transplacental route [10,11].

Question 6. Lines 76-86. Please see the proposed edited language for better clarity and consistency with other published references on maternal and congenital syphilis terminology. 

“According to the Brazil MoH, women diagnosed with syphilis during pregnancy, at delivery and / or puerperium should be reported as SiP. This case definition includes symptomatic or asymptomatic pregnant women with at least one reactive syphilis test, either treponemal or nontreponemal (of any titer), and without previous recorded syphilis treatment. The case definition of CS includes: all newborns, stillbirths or abortions from women diagnosed with syphilis that have not been treated or who received inadequate treatment, children under 13 years of age with clinical manifestations, radiographic or radiological alterations and reactive nontreponemal or treponemal syphilis tests, and children, products of abortion or stillbirth with biopsy or necropsy microbiological evidence of T. pallidum infection in a sample of nasal discharge or skin lesion, detection of T. pallidum by means of direct exams by microscopy (dark field or with colored material) [13].”

Reply: We thank the reviewer for this valuable contribution. The text has been clarified and we have changed the sentence as suggested:

Before: “According to the MoH, it is defined that all cases of women diagnosed with syphilis during prenatal, delivery and / or puerperium should be reported as SiP. Being symptomatic or asymptomatic, it is necessary to present at least one reagent test, being treponemic and / or non-treponemic with any titration and without previous treatment record. For SC, it is considered a case, every newborn, stillborn or abortion of a woman with syphilis that is not treated or treated in an inadequate way; every child under 13 years of age with clinical manifestations, radiographic or radiological alterations and non-treponemic or treponemic reagent tests; microbiological evidence of T. pallidum infection in a sample of nasal discharge or skin lesion, biopsy or necropsy of a child, abortion or stillbirth; detection of T. pallidum by means of direct exams by microscopy (dark field or with colored material) [13]’’.

After (line 82-93): “According to the Brazil MoH, women diagnosed with syphilis during pregnancy, at delivery and / or puerperium should be reported as SiP. This case definition includes symptomatic or asymptomatic pregnant women with at least one reactive syphilis test, either treponemal or nontreponemal (of any titer), and without previous recorded syphilis treatment. The case definition of CS includes: all newborns, stillbirths or abortions from women diagnosed with syphilis that have not been treated or who received inadequate treatment, children under 13 years of age with clinical manifestations, radiographic or radiological alterations and reactive nontreponemal or treponemal syphilis tests, and children, products of abortion or stillbirth with biopsy or necropsy microbiological evidence of T. pallidum infection in a sample of nasal discharge or skin lesion, detection of T. pallidum by means of direct exams by microscopy (dark field or with colored material) [14]’’.

Question 7. Lines 88-89. Please provide appropriate references for the statements regarding increases in syphilis. Reference 14 is from 2015 and likely reflects data from several years prior. Consider using the WHO global estimates of syphilis (Rowley et al. described earlier) and the US CDC 2019 surveillance report https://www.cdc.gov/std/statistics/2019/default.htm for the US as current references. 

Reply: We thank the reviewer for this valuable contribution. We have only added Rowley's reference, which provides more general and up-to-date data.

Before: “Globally, rising numbers of syphilis cases have also been reported in the United States, Canada, Europe, Russia and China…”

After (line 94-96): “Globally, an increasing number of syphilis cases have also been reported in Africa, the Americas, the Eastern Mediterranean, Europe, Southeast Asia, and the Western Pacific [2,3] […]’’

Question 8. Methods: please describe in more detail the joinpoint method of identifying the year intervals of analysis.

Reply: Thank the reviewer for his/her comment. We have inserted the following sentence in the Statistical analysis subsection:

Before: “…Temporal changes in annual incidence rates were calculated using the joinpoint regression model and expressed as Annual Percentage Change (APC) with 5% significance (p < 0.05). Digital maps were obtained from…”

After (line 162-176): “Temporal changes in annual incidence rates were calculated using the joinpoint regression model and expressed as Annual Percentage Change (APC) with 5% significance (p < 0.05) using the NCI Joinpoint regression program version 4.1.1 [17,18]. To determine the optimal number of joinpoints, sequential permutation tests were performed during model selection. Each of the permutation tests performs a test of the null hypothesis H0: number of joinpoints = ka against the alternative Ha: number of joinpoints = kb with ka < kb. The procedure starts with ka = MIN or minimum number of joinpoints and kb = MAX or maximum number of joinpoints, in our case 0 and 5, respectively. Monte Carlo simulation, with the number of permutations fixed at 4,499, is used to calculate the permutation p-value for each hypothesis test. Based on the Joinpoint Regression Program recommendations for the number of time points of observations in our study, our analyzes allowed for a maximum of five joinpoints, meaning that between one and six trend segments could be included in the final model, depending on the number of joinpoints detected [19]’’.

Question 9. What does a skin color of ‘yellow’ imply? The skin colors are difficult to interpret without the addition of information related to culture, origin, or ethnicity. (Asian, Hispanic, Aboriginal, African etc). “Mixed-race” is used but it is unclear how the other races can be derived from skin color alone. If alternatives or co-naming (e.g. yellow = Asian decent) are not possible would explain the historical context of this color-based naming.

Reply: We agree with the reviewer and have revised the shelf-reposted skin color throughout the manuscript and in the legends of Figures 3 and 4. We have also added a sentence at the end of the Study Design subsection that reads as follows. Indeed, yellow = Asian ancestry; white = European ancestry.

After: no text.

Before (line 147-150): Self-reported skin color was classified as European ancestry, dark skin, East Asian ancestry, indigenous ancestry, mixed-race (persons whose skin color is not classified as dark-skinned, European, indigenous, or East Asian).

Question 10. Discussion: line 292. The use of “tendency” is unclear. Consider rate of case number.

Reply: Thank you for the suggestion to change the term used.

Before: “… with an increasing tendency in case numbers over time…”

After (line 316): … an increasing in case numbers over time…

Question 11. Discussion Lines 296-299. Would consider describing Rede Cegonha later in the discussion alongside penicillin shortages.

Reply: We agree with the change and appreciate the suggestion.

Before: “The high number of cases of CS and SiP observed from 2011 to 2017 could have also been due to the implementation of a governmental program in 2011 called "Rede Cegonha" (stork program), which expanded early detection efforts as a result of periodic screening for prenatal, intrapartum and postpartum syphilis [16]”.

After (line 336-341): It is important to note that changes in the epidemiologic infection profile in recent years are likely associated with a) increased testing coverage enabled by the adoption of rapid diagnostic tests, which therefore allowed for the expanded identification of incident cases throughout the country [24–26], b) the shortage of benzathine penicillin, which, since 2014, has affected Brazil, as well as other countries, due to a deficit of raw materials required for its production [27], and c) implementation of a governmental program in 2011 called "Rede Cegonha" (stork program), which expanded early detection efforts as a result of periodic screening for prenatal, intrapartum and postpartum syphilis [28].

Question 12. Discussion line 295: instead of “syphilis remains a national public health concern” consider that “syphilis should be “prioritized” as a national public health concern due to the dramatic increases in rates of SiP and CS”. This would be a good ending for the first paragraph.

Reply: We thank the reviewer for this valuable contribution.

Before: “Regardless, the increase in cases witnessed in recent years suggests that syphilis remains a national public health concern.”

After (line 319-321): “Regardless, the increase in cases witnessed in recent years suggests that syphilis should be “prioritized” as a national public health concern due to the dramatic increases in rates of SiP and CS’’.

Question 13. The discussion is quite long and repeats information about gender of babies and race of mothers. Would recommend shortening and removing redundancy.

Reply: We appreciate the suggestion. Possible changes have been made to improve readability.

Before: “Herein, most cases of CS were diagnosed less than seven days after birth, and 70% of mothers reported receiving prenatal care. Most cases of CS are asymptomatic at birth, however, the high coverage of hospital births with a notification system for CS based on maternity services are the main reason why diagnosis usually occurs within seven days after birth (typically between the 1st and 2nd day of life) [31,32]. Contrarily, in addition to prematurity, newborns can present signs and symptoms of infection soon after birth, including low birth weight, anemia, jaundice, respiratory distress, visceromegaly, congenital malformations, serosanguinous discharge and rhinitis, skin lesions, heart disease and/or hearing loss [33]. Interestingly, it was found that despite the predominance of prenatal care in 70+% of the CS cases investigated herein, significantly high numbers of cases of CS were nonetheless reported throughout the country […]”.

After (line 357-363): “Herein, most cases of CS were diagnosed in asymptomatic children less than seven days after birth with 70% mothers reported receiving prenatal care. However, the high coverage of hospital births with a notification system for CS based on maternity services are the main reason why diagnosis usually occurs within seven days after birth (typically between the 1st and 2nd day of life) [33,34]. Interestingly, it was found that despite the predominance of prenatal care in 70+% of the CS cases investigated herein, significantly high numbers of cases of CS were nonetheless reported throughout the country”.

Before: “However, despite the expansion of diagnosis and treatment in Brazil, increases in the number of cases indicates shortcomings in the efforts designed to control and prevent this STI [34]. At the same time, health authorities have also attributed increases in incidence to the success of public health actions in improving detection rates. Nevertheless, actions designed to improve health care access for pregnant women have performed poorly in terms of CS prevention [26]. One of the main purposes of prenatal care is to assist women in a qualified and humanized way beginning in the early stages of pregnancy, adopting early screening procedures coupled with timely interventions [35]. Early diagnosis and treatment of SiP, ideally before the 20th week of pregnancy, can reduce CS-related cases, such as miscarriages, stillbirths, and infant deaths [36]. It is therefore important that all pregnant women be tested at the first prenatal visit scheduled in the first trimester of pregnancy, with repeat testing performed at around 28 weeks (beginning of the third trimester) and on admission to childbirth in order to promptly implement appropriate therapy if necessary [37-40]. Information regarding CS vertical transmission should be provided to pregnant women at the onset of prenatal care, and physicians must inform patients regarding the risks and consequences of the disease to the mother and her fetus [30,41] ….. An established diagnosis does not guarantee adhesion to appropriate treatment by the patient. The late onset of symptoms (mostly detected in the third trimester), the interruption of and/or low attendance in prenatal examinations, difficulties in diagnosis, a lack of information regarding infection and unsafe sexual practices have been reported as risk factors for the development of syphilis [42–44]. Furthermore, errors in antibiotic dosage have also been of concern. Other key factors, such as conjugal infidelity, the absence of partners at prenatal appointments and reluctance in adhering to treatment protocols have also been reported by patients [41,45]’’.

After (line 368-383): “Despite the expansion of diagnosis and treatment in Brazil, increases in the number of cases indicates shortcomings in the efforts designed to control and prevent this STI [36]. At the same time, health authorities have also attributed increases in incidence to the success of public health actions in improving detection rates. Nevertheless, actions designed to improve health care access for pregnant women have performed poorly in terms of CS prevention [25]. Early diagnosis and treatment of SiP can reduce CS-related cases, such as miscarriages, stillbirths, and infant deaths [37]. Therefore, it is important that all pregnant women be tested at the first prenatal visit scheduled (1st trimester), with 28 weeks pregnant and at the time of delivery in order to promptly implement appropriate therapy if necessary [32,38–42]. Information regarding CS vertical transmission should be provided to pregnant women at the onset of prenatal care, and physicians must inform patients regarding the risks and consequences of the disease to the mother and her fetus [32,42]. Nonetheless, difficulties in diagnosis, interruption of and/or low attendance in prenatal examinations, non-adherence or inadequate treatment regarding antibiotic dosage, and absence of partners at prenatal appointments have been reported as risk factors for the development and non-interruption of syphilis [42–47].”

Before: “Regarding the sociodemographic profile of pregnant women observed herein, most cases were identified among women aged 20-39 years who self-identified as ‘mixed-race’, had up to eight years of formal education and were mainly diagnosed with primary syphilis. However, it worth noting that many records contained missing self-reported skin color classification data in P1, which can be considered as a bias in the interpretation of our results. Indeed, the data presented herein corroborate other studies that identified a significant correlation between these sociodemographic characteristics and SiP [26,29–32,34,36,42-51]. STI and SiP were found to be more strongly associated with women who dropped out of school, self-identified as 'black' or 'mixed-race', age below 20 years or between 20 and 30 years old [46], without access to quality health services, preventive and educational programs or received assistance at public health care units, including adequate prenatal care [32,43,44]. We suggest that distinct strategies are required to reach more vulnerable populations and to minimize inequalities that enable greater access to health services. Poverty prompts specific vulnerabilities, whether behavioral or brought on by deficiencies in health services, such as prenatal care access and quality, which are also significantly associated with SiP. Brazilian social inequality in health supports the hypothesis that the prevalence of SiP is associated with a lower socioeconomic status [32,43,44].”

After (line 384-394): “Regarding the sociodemographic profile of pregnant women observed herein, most cases were identified among women aged 20-39 years who self-identified as ‘mixed-race’, had up to eight years of formal education and were mainly diagnosed with primary syphilis. However, it worth noting that many records contained missing self-reported skin color classification data in P1, which can be considered as a bias in the interpretation of our results. Indeed, the data presented herein corroborate other studies [25,31–34,36,37,43–46,48–53] that identified a significant correlation between SiP and women who dropped out of school, self-identified as 'black' or 'mixed-race', age below 20 years or between 20 and 30 years old [48], without access to quality health services, preventive and educational programs or received assistance at public health care units, including adequate prenatal care [34,44,45,54,55].”

Before: “[…] The observed variations in incidence among the municipalities may be the result of a decline in the underreporting of cases or reflect problems in local health systems, such as a lack of access to specialized services. Importantly, incomplete reporting hinders the elaboration of preventive strategies by policymakers, resulting in ineffective epidemiological surveillance [51,52]. It is evident that the Brazilian healthcare system will continue to be challenged by this scenario, as despite government investment in awareness campaigns, the circumstances remain far from ideal. Low adherence to treatment among patients and their partners is a main obstacle that must be overcome. Insufficient social awareness regarding prevention and treatment reflects the urgent need for educational policies aimed at preventing congenital infections [32,51] in Brazil, especially in the affected macroregions and microregions identified in this study.”

After (line 415-422): “The observed variations in incidence among the municipalities may be the result of a decline in the underreporting of cases or reflect problems in local health systems, such as a lack of access to specialized services. Importantly, incomplete reporting hinders the elaboration of preventive strategies by policymakers, resulting in ineffective epidemiological surveillance [53,63]. It is evident that the Brazilian healthcare system will continue to be challenged by this scenario, as despite government investment in awareness campaigns, the circumstances remain far from ideal.”

Before: “Low adherence to treatment among patients and their partners is a main obstacle that must be overcome. Insufficient social awareness regarding prevention and treatment reflects the urgent need for educational policies aimed at preventing congenital infections [32,51] in Brazil, especially in the affected macroregions and microregions identified in this study […]

We conclude that despite the existence of control and awareness programs for STIs […]”

After (line 423-432): “We suggest that distinct strategies are required to reach more vulnerable populations and to minimize inequalities that enable greater access to health services. Poverty prompts specific vulnerabilities, whether behavioral or brought on by deficiencies in health services, such as prenatal care access and quality, which are also significantly associated with SiP. Brazilian social inequality in health supports the hypothesis that the prevalence of SiP is associated with a lower socioeconomic status [34,44,45]. SiP control programs should place greater focus on these more vulnerable populations [44,45], especially considering that the lack of or inadequacies in public sexual education policies for younger individuals was associated with decreased condoms use in casual sexual relations in recent years [34] […]

(Line 442): In this way, concluded despite the existence of control and awareness programs for […]”

Question 14. Line 315 would reference global estimates of CS by Korenromp et al stated comment number 4.

Reply: We have changed the reference according to the reviewer's suggestion.

Question 15. Line 320 needs a reference for global BPG shortages. Suggest this reference which highlights Brazil. Nurse-Findlay S, et al Françoise Bigirimana F, Ouedraogo L, Pyne-Mercier L. Supply, Demand, and Shortages of Benzathine Penicillin for Treatment of Syphilis: A Market Assessment. PLoS Medicine. 2017;14 (12):e1002473.

Reply: We have included the reference according to the reviewer's suggestion.

Question 16. Line 345 would consider the Brazilian reference Rocha AFB et al. Complications, clinical manifestations of congenital syphilis and aspects related to its prevention: an integrative review. https://pubmed.ncbi.nlm.nih.gov/34287560/.

Reply: We have changed the reference according to the reviewer's suggestion.

Question 14. Discussion lines 326 -321 from “A similar prevalence… “Would delete this phrase as it is well known that the syphilis does not preferentially affect the gender of the infant.

Reply: We thank the reviewer for his/her criticism and politely ask him/her to reconsider this opinion. The results presented and the discussion confirm even more the observations made and therefore we believe that it is important to highlight them in the study, since these data are provided by the Ministry of Health and were included in the analysis along with the other variables.

Question 18. Recommend other areas to shorten for example: “With respect to self-reported skin color, the majority of reported CS cases occurred among newborns identified by the family as ‘mixed-race’. Other data have also shown that the incidence of congenital syphilis tends to be significantly higher in black or ‘mixed-race’ children. In fact, several studies have linked cases of CS with family history, including the racial classification of children’s parents, with high numbers of pregnant women self-identifying as ‘mixed-race’ or black [22–25].”……….Can be shortened to: “The majority of reported CS cases occurred among newborns identified by the family as ‘mixed-race’ similar to other reports [22-25]”. Shortening the discussion to the relevant details will ensure more people read it.

Reply: We thank the reviewer for this valuable contribution. We have changed the sentence according to the reviewer's suggestion.

Before: “With respect to self-reported skin color, the majority of reported CS cases occurred among newborns identified by the family as ‘mixed-race’. Other data have also shown that the incidence of congenital syphilis tends to be significantly higher in black or ‘mixed-race’ children. In fact, several studies have linked cases of CS with family history, including the racial classification of children’s parents, with high numbers of pregnant women self-identifying as ‘mixed-race’ or black [22–25].”

After: (line 355-355): The majority of reported CS cases occurred among newborns identified by the family as ‘mixed-race’ similar to other reports [31–34].

Question 19. Citation 32 would change to the global guidelines https://www.who.int/reproductivehealth/publications/rtis/syphilis-ANC-screenandtreat-guidelines/en/.

Reply: The reference has been updated as suggested.

Before: 32. Newman L, Kamb M, Hawkes S, Gomez G, Say L, Seuc A, et al. Global estimates of syphilis in pregnancy and associated adverse outcomes: analysis of multinational antenatal surveillance data. PLoS Med. 2013;10: e1001396. doi:10.1371/journal.pmed.1001396”.

Reference # 39 (line 570-573): World Health Organization. WHO guideline on syphilis screening and treatment for pregnant women. World Health Organization, 2017.

Question 20. The paragraphs starting on lines 380 and 394 are nearly identical in content and interpretation. Would delete one of these paragraphs. There are multiple examples of this type of duplication in the discussion. As much as possible, would recommend that the authors stay close to the findings reflected from their spatiotemporal analysis with aa greatly shortened summary of the literature that contributes to the understanding of risk factors and other contributors to increasing syphilis diagnoses in Brazil.

Reply: We appreciate the suggestion. We deleted what was repeated and joined the information in a single paragraph.

Before: “Regarding the sociodemographic profile of pregnant women observed herein, most cases were identified among women aged 20-39 years who self-identified as ‘mixed-race’, had up to eight years of formal education and were mainly diagnosed with primary syphilis…” 

“Several child and maternal factors have been associated with increased risk and vulnerability to CS, such as self-reported skin color, socioeconomic status and maternal age [24]. In fact, STI and SiP were found to be more strongly associated with women who dropped out of school, self-identified as 'black' or 'mixed-race', were under 20 years of age or between 20 and 30 years old [41], had limited access to quality health services, preventive and educational programs or received assistance at public health care units, but without adequate prenatal care [25,35,36].”

After (line 384-394): “Regarding the sociodemographic profile of pregnant women observed herein, most cases were identified among women aged 20-39 years who self-identified as ‘mixed-race’, had up to eight years of formal education and were mainly diagnosed with primary syphilis. However, it worth noting that many records contained missing self-reported skin color classification data in P1, which can be considered as a bias in the interpretation of our results. Indeed, the data presented herein corroborate other studies [25,31–34,36,37,43–46,48–53] that identified a significant correlation between SiP and women who dropped out of school, self-identified as 'black' or 'mixed-race', age below 20 years or between 20 and 30 years old [48], without access to quality health services, preventive and educational programs or received assistance at public health care units, including adequate prenatal care [34,44,45,54,55].”

 

Author's Reply to the Review Report (Reviewer 3)

The manuscript PONE-D-21-31051, entitled "Spatiotemporal distribution analysis of syphilis in Brazil: Cases of congenital and syphilis in pregnant women from 2001-2017" is an interesting paper and highlights an important problem for public health in Brazil. I have few comments and suggestions:

Question 1. Pag. 3, lines 56-58: Brazilian data in the first paragraph should be updated. The last data are from 2020.

Reply: Thank you for the suggestion to update the information. We have made the suggested correction by changing the reference.

Before: “Increasing incidence and prevalence has been reported since 2008 in adults between 15 and 49 years of age, being 10.6 million cases of syphilis”.

After (line 54-57): “According to the number of reported cases of curable STIs in 2016, only 6.3 million (1.67%) were syphilis in women and men aged 15-49 years. Looking at the period from 2012 to 2016, the estimated global prevalence was 0.5% and the incidence was estimated at 1.7 cases per 1,000 women and 1.6 cases per 1,000 men [2]’’.

Question 2. Pag.3, lines 69-70: Even if the paper is about syphilis in pregnancy and congenital syphilis, it should be included that there is an ordinance for syphilis in adults (acquired syphilis). It is an infection of compulsory notification since 2010.

Reply. We welcome the suggestion and agree that acquired syphilis should be included. However, we have had some difficulty obtaining and analyzing these data. Despite the reporting requirement, we were unable to obtain the raw data from SINAN, and the available information is already published in the Epidemiologic Bulletins of the Ministry of Health, in a shorter time period than that examined in this paper. Furthermore, when using this information from the epidemiological bulletins for joinpoint regression analyzes using APC (Annual Percent Change), acquired syphilis did not show distinct periods such as CS and SiP. This result does not confirm the increase in the number of cases in Brazil and could be due to the lack of information in the period proposed by the study.

Question 3. Pag. 5, line 115: Change “Brazilian National Census” to Brazilian National Estimates”. The last census was in 2010.

Reply. We have replaced “Brazilian National Census” with Brazilian National Estimates: 

Before: “According to the 2015 Brazilian national census, the country’s total […]’’

After (line 123): According to the 2015 Brazilian national estimates, the country’s total […]’’

Question 4. Page 9, lines 220-221: 40% of the clinical stage of syphilis were classified as primary. The misclassification is a big problem in Brazil and the authors should emphasize it in the discussion section.

Reply: Good observation. We appreciate and accept the suggestion.

Before: No text.

After (line 394-398): “As for cases of primary syphilis in pregnant women, it is noteworthy that like the misclassification of latent syphilis in the United States [56], the estimated number of 40% found here may also have been misclassified, as this is considered a major problem. in Brazil. In fact, the correct classification of the clinical stage of the infection depends on experienced medical personnel’’.

Question 5. Page 11, lines 252-256: The authors describe data of syphilis in pregnancy during pregnancy. In page 13, lines 300-308, the authors need to highlight in the discussion that the compulsory notification started in 2005 for pregnant women.

Reply. Thanks for the observation.

Before: “Additionally, the detection rate in pregnant women rose from 0.5 to 17.2 cases per 1,000 live births between 2005 and 2017’’.

After (line 324-327): …which indicate an increase from 1.7 to 8.6 cases/1,000 live births for CS between 2003 and 2017 and 0.5 to 17.2 cases per 1,000 live births for SiP between 2005 and 2017, with SiP only becoming reportable in 2005.

Question 6. Page 15, lines 350-355: Are there links to access these documents listed in Portuguese? If not, they could be excluded.

Reply. Access links have been added in the references. Thanks for the suggestion.

Before: “[…] provided to all pregnant women (Brasil. Ministério da Saúde. Secretaria de Atenção à Saúde. Departamento de Ações Programáticas Estratégicas. Área Técnica de Saúde da Mulher 2006). It is known that the incidence of syphilis is considered to be an important indicator of accessibility and prenatal care quality (Ministério da Saúde. Secretaria de Vigilância em Saúde. Programa Nacional de DST/AIDS 2007, Ministério da Saúde. Portaria nº 1.459, de 24 de junho de 2011).”

After (line 366-367): “properly provided to all pregnant women [35]. It is known that the incidence of syphilis is an important indicator of accessibility and prenatal care quality [28].”

Reference # 28 (line 533-537): Ministério da Saúde. Secretaria de Atenção à Saúde. Departamento de Ações Programáticas Estratégicas. Área Técnica de Saúde da Mulher. 2006 [Cited 2022 March 25]. Available: https://bvsms.saude.gov.br/bvs/publicacoes/relatorio_2003a2006_politica_saude_mulher.pdf.

Reference # 35 (line 558-560): Ministério da Saúde. Secretaria de Vigilância em Saúde. Programa Nacional de DST/AIDS 2007. Portaria nº 1.459, de 24 de junho de 2011. Available: https://bvsms.saude.gov.br/bvs/saudelegis/gm/2011/prt1459_24_06_2011.html

Question 7. Page 15, lines 365-368: Authors explain about syphilis tests during pregnancy. It is important to include the recommendation to test for syphilis during labor. Both WHO and Brazilian guidelines recommend it.

Reply Thank you for pointing out this important information.

Before: “It is therefore important that all pregnant women be tested at the first prenatal visit scheduled in the first trimester of pregnancy, with repeat testing performed at around 28 weeks (beginning of the third trimester) in order to promptly implement appropriate therapy if necessary”.

After (line 374-377): Therefore, it is important that all pregnant women be tested at the first prenatal visit scheduled (1st trimester), with 28 weeks pregnant and at the time of delivery to promptly implement appropriate therapy if necessary [32,38–42].

Question 8. The discussion section is a little to long, it should be optimized. There is discussion on demographics, as age, mixed-race, education, in three different parts.

Reply: We appreciate the suggestion. Possible changes have been made to improve readability.

Before: “Herein, most cases of CS were diagnosed less than seven days after birth, and 70% of mothers reported receiving prenatal care. Most cases of CS are asymptomatic at birth, however, the high coverage of hospital births with a notification system for CS based on maternity services are the main reason why diagnosis usually occurs within seven days after birth (typically between the 1st and 2nd day of life) [31,32]. Contrarily, in addition to prematurity, newborns can present signs and symptoms of infection soon after birth, including low birth weight, anemia, jaundice, respiratory distress, visceromegaly, congenital malformations, serosanguinous discharge and rhinitis, skin lesions, heart disease and/or hearing loss [33]. Interestingly, it was found that despite the predominance of prenatal care in 70+% of the CS cases investigated herein, significantly high numbers of cases of CS were nonetheless reported throughout the country […].”

After (line 357-363): “Herein, most cases of CS were diagnosed in asymptomatic children less than seven days after birth with 70% mothers reported receiving prenatal care. However, the high coverage of hospital births with a notification system for CS based on maternity services are the main reason why diagnosis usually occurs within seven days after birth (typically between the 1st and 2nd day of life) [33,34]. Interestingly, it was found that despite the predominance of prenatal care in 70+% of the CS cases investigated herein, significantly high numbers of cases of CS were nonetheless reported throughout the country.”

Before: “However, despite the expansion of diagnosis and treatment in Brazil, increases in the number of cases indicates shortcomings in the efforts designed to control and prevent this STI [34]. At the same time, health authorities have also attributed increases in incidence to the success of public health actions in improving detection rates. Nevertheless, actions designed to improve health care access for pregnant women have performed poorly in terms of CS prevention [26]. One of the main purposes of prenatal care is to assist women in a qualified and humanized way beginning in the early stages of pregnancy, adopting early screening procedures coupled with timely interventions [35]. Early diagnosis and treatment of SiP, ideally before the 20th week of pregnancy, can reduce CS-related cases, such as miscarriages, stillbirths, and infant deaths [36]. It is therefore important that all pregnant women be tested at the first prenatal visit scheduled in the first trimester of pregnancy, with repeat testing performed at around 28 weeks (beginning of the third trimester) and on admission to childbirth in order to promptly implement appropriate therapy if necessary [37-40]. Information regarding CS vertical transmission should be provided to pregnant women at the onset of prenatal care, and physicians must inform patients regarding the risks and consequences of the disease to the mother and her fetus [30,41].

An established diagnosis does not guarantee adhesion to appropriate treatment by the patient. The late onset of symptoms (mostly detected in the third trimester), the interruption of and/or low attendance in prenatal examinations, difficulties in diagnosis, a lack of information regarding infection and unsafe sexual practices have been reported as risk factors for the development of syphilis [42–44]. Furthermore, errors in antibiotic dosage have also been of concern. Other key factors, such as conjugal infidelity, the absence of partners at prenatal appointments and reluctance in adhering to treatment protocols have also been reported by patients [41,45].”

After (line 368-383): “Despite the expansion of diagnosis and treatment in Brazil, increases in the number of cases indicates shortcomings in the efforts designed to control and prevent this STI [36]. At the same time, health authorities have also attributed increases in incidence to the success of public health actions in improving detection rates. Nevertheless, actions designed to improve health care access for pregnant women have performed poorly in terms of CS prevention [25]. Early diagnosis and treatment of SiP can reduce CS-related cases, such as miscarriages, stillbirths, and infant deaths [37]. Therefore, it is important that all pregnant women be tested at the first prenatal visit scheduled (1st trimester), with 28 weeks pregnant and at the time of delivery in order to promptly implement appropriate therapy if necessary [32,38–42]. Information regarding CS vertical transmission should be provided to pregnant women at the onset of prenatal care, and physicians must inform patients regarding the risks and consequences of the disease to the mother and her fetus [32,42]. Nonetheless, difficulties in diagnosis, interruption of and/or low attendance in prenatal examinations, non-adherence or inadequate treatment regarding antibiotic dosage, and absence of partners at prenatal appointments have been reported as risk factors for the development and non-interruption of syphilis [42–47].”

Before: “Regarding the sociodemographic profile of pregnant women observed herein, most cases were identified among women aged 20-39 years who self-identified as ‘mixed-race’, had up to eight years of formal education and were mainly diagnosed with primary syphilis. However, it worth noting that many records contained missing self-reported skin color classification data in P1, which can be considered as a bias in the interpretation of our results. Indeed, the data presented herein corroborate other studies that identified a significant correlation between these sociodemographic characteristics and SiP [26,29–32,34,36,42-51]. STI and SiP were found to be more strongly associated with women who dropped out of school, self-identified as 'black' or 'mixed-race', age below 20 years or between 20 and 30 years old [46], without access to quality health services, preventive and educational programs or received assistance at public health care units, including adequate prenatal care [32,43,44]. We suggest that distinct strategies are required to reach more vulnerable populations and to minimize inequalities that enable greater access to health services. Poverty prompts specific vulnerabilities, whether behavioral or brought on by deficiencies in health services, such as prenatal care access and quality, which are also significantly associated with SiP. Brazilian social inequality in health supports the hypothesis that the prevalence of SiP is associated with a lower socioeconomic status [32,43,44].”

After (line 384-394): Regarding the sociodemographic profile of pregnant women observed herein, most cases were identified among women aged 20-39 years who self-identified as ‘mixed-race’, had up to eight years of formal education and were mainly diagnosed with primary syphilis. However, it worth noting that many records contained missing self-reported skin color classification data in P1, which can be considered as a bias in the interpretation of our results. Indeed, the data presented herein corroborate other studies [25,31–34,36,37,43–46,48–53] that identified a significant correlation between SiP and women who dropped out of school, self-identified as 'black' or 'mixed-race', age below 20 years or between 20 and 30 years old [48], without access to quality health services, preventive and educational programs or received assistance at public health care units, including adequate prenatal care [34,44,45,54,55]. 

Before: “[…] The observed variations in incidence among the municipalities may be the result of a decline in the underreporting of cases or reflect problems in local health systems, such as a lack of access to specialized services. Importantly, incomplete reporting hinders the elaboration of preventive strategies by policymakers, resulting in ineffective epidemiological surveillance [51,52]. It is evident that the Brazilian healthcare system will continue to be challenged by this scenario, as despite government investment in awareness campaigns, the circumstances remain far from ideal. Low adherence to treatment among patients and their partners is a main obstacle that must be overcome. Insufficient social awareness regarding prevention and treatment reflects the urgent need for educational policies aimed at preventing congenital infections [32,51] in Brazil, especially in the affected macroregions and microregions identified in this study.”

After (line 415-422): “The observed variations in incidence among the municipalities may be the result of a decline in the underreporting of cases or reflect problems in local health systems, such as a lack of access to specialized services. Importantly, incomplete reporting hinders the elaboration of preventive strategies by policymakers, resulting in ineffective epidemiological surveillance [53,62]. It is evident that the Brazilian healthcare system will continue to be challenged by this scenario, as despite government investment in awareness campaigns, the circumstances remain far from ideal.”

Before: “Low adherence to treatment among patients and their partners is a main obstacle that must be overcome. Insufficient social awareness regarding prevention and treatment reflects the urgent need for educational policies aimed at preventing congenital infections [32,51] in Brazil , especially in the affected macroregions and microregions identified in this study […]

We conclude that despite the existence of control and awareness programs for STIs […]”

After (line 423-432): We suggest that distinct strategies are required to reach more vulnerable populations and to minimize inequalities that enable greater access to health services. Poverty prompts specific vulnerabilities, whether behavioral or brought on by deficiencies in health services, such as prenatal care access and quality, which are also significantly associated with SiP. Brazilian social inequality in health supports the hypothesis that the prevalence of SiP is associated with a lower socioeconomic status [34,44,45]. SiP control programs should place greater focus on these more vulnerable populations [44,45], especially considering that the lack of or inadequacies in public sexual education policies for younger individuals was associated with decreased condoms use in casual sexual relations in recent years [34].

(Line 441): In this way, concluded despite the existence of control and awareness programs for […]

Question 9. I did not see in the discussion section a discussion about the use of secondary data. It is important to report it.

Reply: We thank the reviewer for the suggestion.

Before: “We performed a systematic spatiotemporal analysis of reported cases of CS and SiP in Brazil…’’

After (line 314-315): We performed a systematic spatiotemporal analysis using secondary data of reported cases of CS and SiP in Brazil…

Question 10. References should be revisited because they should follow the journal recommendations. There are different presentation of the number of authors and one of them are in caps letter.

Reply: We thank the reviewer for calling this to our attention. Changes have been made.

---

## [Decision Letter · Decision Letter 1]

4 Sep 2022

PONE-D-21-31051R1Spatiotemporal distribution analysis of syphilis in Brazil: Cases of congenital and syphilis in pregnant women from 2001-2017PLOS ONE

Dear Dr. Santos,

Thank you for submitting your manuscript to PLOS ONE. After careful consideration, we feel that it has merit but does not fully meet PLOS ONE’s publication criteria as it currently stands. Therefore, we invite you to submit a revised version of the manuscript that addresses the points raised during the review process.

We look forward to receiving your revised manuscript.

Kind regards,

Antonio Simone Laganà, M.D., Ph.D.

Academic Editor

PLOS ONE

Journal Requirements:

Additional Editor Comments:

One of the reviewers has still some minor concerns.

For this reason, I invite authors to perform these minor corrections (as recommended by the reviewer) before to consider the manuscript for final acceptance.

Reviewers' comments:

Reviewer's Responses to Questions

**Comments to the Author**

1. If the authors have adequately addressed your comments raised in a previous round of review and you feel that this manuscript is now acceptable for publication, you may indicate that here to bypass the “Comments to the Author” section, enter your conflict of interest statement in the “Confidential to Editor” section, and submit your "Accept" recommendation.

Reviewer #1: (No Response)

2. Is the manuscript technically sound, and do the data support the conclusions?

Reviewer #1: Yes

3. Has the statistical analysis been performed appropriately and rigorously? 

Reviewer #1: Yes

4. Have the authors made all data underlying the findings in their manuscript fully available?

Reviewer #1: Yes

5. Is the manuscript presented in an intelligible fashion and written in standard English?

Reviewer #1: Yes

6. Review Comments to the Author

Reviewer #1: I am overall happy with the changes and the improved manuscript. I have a minor (yet very important) comment with regards to the graphs presented in the appendix of the manuscript.

Fig. 3: The 'yellow' category in the legend has been replaced with a much more appropriate 'Asian ancestry'. However, the graph title is 'self reported skin color', a category that has been outdated for a while. Self reported race or ethnicity is more appropriate than skin color.

Fig. 4: still includes 'yellow' in the legend.

7. PLOS authors have the option to publish the peer review history of their article (what does this mean?). If published, this will include your full peer review and any attached files.

Reviewer #1: No

---

## [Author Response · Author response to Decision Letter 1]

6 Sep 2022

Dear Dr. Laganà,

Editor Plos One

We thank you and the reviewers again for your thoughtful suggestions that helped improve our manuscript. We believe that this revised manuscript is a better and more balanced representation of our research, and we hope that it is now suitable for publication in your journal. The answers to the questions can be found below (PONE-D-21-31051R2).

Question 1 - Reviewer #1. Fig. 3: The 'yellow' category in the legend has been replaced with a much more appropriate 'Asian ancestry'. However, the graph title is 'self reported skin color', a category that has been outdated for a while. Self reported race or ethnicity is more appropriate than skin color.

Reply: We thank the reviewer for bringing this error to our attention. We have updated the graph title and changed the Fig. 3.

Question 2 - Reviewer #1. Fig. 4: still includes 'yellow' in the legend.

Reply: We thank the reviewer for this observation We have updated the Fig. 4.

---

## [Editor Report · Decision Letter 2]

22 Sep 2022

Spatiotemporal distribution analysis of syphilis in Brazil: Cases of congenital and syphilis in pregnant women from 2001-2017

PONE-D-21-31051R2

Dear Dr. Santos,

We’re pleased to inform you that your manuscript has been judged scientifically suitable for publication and will be formally accepted for publication once it meets all outstanding technical requirements.

Kind regards,

Antonio Simone Laganà, M.D., Ph.D.

Academic Editor

PLOS ONE

Additional Editor Comments (optional):

I carefully evaluated the revised version of this manuscript.

Authors have performed the required changes, improving significantly the quality of the paper.
---

## [Editor Report · Acceptance letter]

27 Sep 2022

PONE-D-21-31051R2 

Spatiotemporal distribution analysis of syphilis in Brazil: Cases of congenital and syphilis in pregnant women from 2001-2017 

Dear Dr. Santos:

I'm pleased to inform you that your manuscript has been deemed suitable for publication in PLOS ONE. Congratulations! Your manuscript is now with our production department. 

Kind regards, 

on behalf of

Dr. Antonio Simone Laganà 

Academic Editor

PLOS ONE